# PROBABILISTIC KERNEL FUNCTION FOR FAST ANGLE TESTING

Kejing Lu[1], Chuan Xiao[2,3], and Yoshiharu Ishikawa[3]

[1]Yamanashi University, k.riku@yamanashi.ac.jp
[2]Osaka University, chuanx@ist.osaka-u.ac.jp
[3]Nagoya University, ishikawa@i.nagoya-u.ac.jp

## ABSTRACT

In this paper, we study the angle testing problem in the context of similarity search in high-dimensional Euclidean spaces and propose two projection-based probabilistic kernel functions, one designed for angle comparison and the other for angle thresholding. Unlike existing approaches that rely on random projection vectors drawn from Gaussian distributions, our approach leverages reference angles and adopts a deterministic structure for the projection vectors. Notably, our kernel functions do not require asymptotic assumptions, such as the number of projection vectors tending to infinity, and can be theoretically and experimentally shown to outperform Gaussian-distribution-based kernel functions. We apply the proposed kernel function to Approximate Nearest Neighbor Search (ANNS) and demonstrate that our approach achieves a $2.5\times$–$3\times$ higher query-per-second (QPS) throughput compared to the widely-used graph-based search algorithm HNSW. Our code and data are available at https://github.com/KejingLu-810/KS.

## 1 INTRODUCTION

Vector-based similarity search is a core problem with broad applications in machine learning, data mining, and information retrieval. It involves retrieving data points in a high-dimensional space that are most similar to a given query vector based on a specific similarity measure. This task is central to many downstream applications, including nearest neighbor classification, recommendation systems, clustering, anomaly detection, and retrieval-augmented generation (RAG). However, the high dimensionality of modern datasets makes efficient similarity search particularly challenging, highlighting the need for fast and scalable vector computation techniques.

Among the various similarity measures for high-dimensional vectors, the $\ell_2$ norm, cosine similarity, and inner product are the most commonly used in practice. As discussed in Yan et al. (2018); Dai et al. (2020); Lu et al. (2024), it is often possible to pre-compute and store the norms of vectors in advance, allowing these measures to be reduced to the computation of the cosine of the angle between two normalized vectors, thereby highlighting the central role of angle computation. On the other hand, in many real-world scenarios, we are not concerned with the exact values of the angles but rather with the outcome—which one is greater—of an angle comparison, which is referred to as angle testing: Given a query vector $q$ and data vectors $v_1, v_2, v$ on the sphere $\mathbb{S}^{d-1}$, typical operations include comparing $\langle q, v_1 \rangle$ and $\langle q, v_2 \rangle$, or determining whether $\langle q, v \rangle$ exceeds a certain threshold. These operations, however, require computing exact cosines of angles, which has a cost of $O(d)$ per comparison and becomes expensive in high dimensions. To address this, we aim to design a computationally efficient probabilistic kernel function $K$ that can approximate these comparisons with reduced cost and high success probability. More precisely, we focus on the following two problems:

**Problem 1.1.** *(**Probabilistic kernel function for comparison**) Design a probabilistic kernel function $K: \mathbb{S}^{d-1} \times \mathbb{S}^{d-1} \to RV$ with computational cost $o(d)$, where $RV$ denotes the set of random variables, such that, for any data vectors $v_1, v_2 \in \mathbb{S}^{d-1}$ and query $q \in \mathbb{S}^{d-1}$ satisfying $\langle q, v_1 \rangle > \langle q, v_2 \rangle$, we have $\Pr[K(q, v_1) > K(q, v_2)] > 1 - \epsilon$, where $\epsilon \leq 0.5$.*

**Problem 1.2.** *(**Probabilistic kernel function for thresholding**) Given a fixed angle threshold $\theta \in (0, \pi)$, design a probabilistic kernel function $K: \mathbb{S}^{d-1} \times \mathbb{S}^{d-1} \to RV$ with computational cost $o(d)$ such that*

*for any $\boldsymbol{q}, \boldsymbol{v_1}, \boldsymbol{v_2} \in \mathbb{S}^{d-1}$ with angles $\phi_1 < \theta$ between $\boldsymbol{q}$ and $\boldsymbol{v_1}$, and $\phi_2 > \theta$ between $\boldsymbol{q}$ and $\boldsymbol{v_2}$, we have $\Pr[K(\boldsymbol{q}, \boldsymbol{v_1}) > \cos\theta] \geq 1 - \epsilon_1$, and $\Pr[K(\boldsymbol{q}, \boldsymbol{v_2}) > \cos\theta] < \epsilon_2$, where $\epsilon_1, \epsilon_2 \leq 0.5$.*

One of the main goals of this paper is to design appropriate $K$'s to solve the above two problems. Before proceeding, we note that much work has focused on the estimation of $\langle \boldsymbol{q}, \boldsymbol{v} \rangle$, such as Johnson-Lindenstrauss (JL) bound (Johnson et al., 1984) and quantization-based techniques (Jégou et al., 2011; Ge et al., 2014; Martinez et al., 2018; Gao & Long, 2024; Guo et al., 2020). In contrast, the defined problems focus on comparing $\langle \boldsymbol{q}, \boldsymbol{v} \rangle$ with another inner product or a threshold, necessitating a distinct theoretical analysis centered on probabilistic decision-making. Beyond the theoretical perspective, these two problems also give rise to a range of practical applications. The goal of Problem 1.1 aligns with that of the random-projection-based technique CEOs (Pham, 2021) under cosine similarity (we postpone the general case of inner product to Sec. 6), allowing the designed kernel function to be applied to tasks where CEOs is effective, such as Maximum Inner Product Search (MIPS) (Pham, 2021), filtering of NN candidates (Pham & Liu, 2022), DBSCAN (Xu & Pham, 2024), and more. Notably, the goal of Problem 1.2 is similar to that of the graph-based ANNS approach PEOs (Lu et al., 2024), making the corresponding kernel function well suited for probabilistic routing tests in similarity graphs, which have demonstrated significant performance improvements over baseline graphs such as HNSW (Malkov & Yashunin, 2020). In Sec. 6, we will further elaborate on the applications of these two probabilistic kernel functions.

Despite addressing different tasks, all the techniques (Pham, 2021; Pham & Liu, 2022; Xu & Pham, 2024; Lu et al., 2024) mentioned above use Gaussian distribution to generate projection vectors and are built upon a common statistical result, as follows.

**Lemma 1.3.** *(Theorem 1 in Pham (2021)) Given two vectors $\boldsymbol{v}, \boldsymbol{q}$ on $\mathbb{S}^{d-1}$, and $m$ random vectors $\{\boldsymbol{u_i}\}_{i=1}^m \sim \mathcal{N}(0, I^d)$, let $\boldsymbol{u}_{\max} = \operatorname{argmax}_{\boldsymbol{u_i}} |\boldsymbol{q}^\top \boldsymbol{u_i}|$. As $m$ goes to infinity, we have:*

$$\boldsymbol{v}^\top \boldsymbol{u}_{\max} \sim \mathcal{N}(\operatorname{sgn}(\boldsymbol{q}^\top \boldsymbol{u}_{\max}) \cdot \boldsymbol{q}^\top \boldsymbol{v} \sqrt{2 \ln m}, 1 - (\boldsymbol{q}^\top \boldsymbol{v})^2). \tag{1}$$

Lemma 1.3 builds the relationship between angles and corresponding projection vectors. Actually, $\boldsymbol{v}^\top \boldsymbol{u}_{\max}$ can be viewed as an indicator of the cosine of the angle between $\boldsymbol{q}$ and $\boldsymbol{v}$. More specifically, the larger $\boldsymbol{v}^\top \boldsymbol{u}_{\max}$ is, the more likely it is that $\boldsymbol{q}^\top \boldsymbol{v}$ is large. On the other hand, all $\boldsymbol{v}^\top \boldsymbol{u_i}$'s can be computed beforehand during the indexing phase and $\boldsymbol{v}^\top \boldsymbol{u}_{\max}$ can be easily accessed during the query phase, making $K(\boldsymbol{q}, \boldsymbol{v}) = \boldsymbol{v}^\top \boldsymbol{u}_{\max}$ a suitable kernel function for various angle testing problems, e.g., Problem 1.1.

However, Lemma 1.3 has a significant theoretical limitation: Relationship (1) relies on the assumption that the number of projection vectors $m$ tends to infinity. Since the computational cost of evaluating projection vectors grows with $m$, $m$ cannot be very large in practice. Moreover, since Pham & Liu (2022); Xu & Pham (2024); Lu et al. (2024) all used Lemma 1.3 to derive their theoretical results, these results are also affected by this limitation, and the impact of $m$ becomes even harder to predict in these applications.

The starting point of our research is to overcome this limitation, and we make the following two key observations. (1) The Gaussian distribution used in Lemma 1.3 is not essential. Instead, the only factor determining the estimation accuracy of $\boldsymbol{q}^\top \boldsymbol{v}$ is the reference angle, that is, the angle between $\boldsymbol{q}$ and $\boldsymbol{u}_{\max}$. (2) By introducing a random rotation matrix, the reference angle becomes dependent on the structure of the projection vectors and is predictable.

Based on these two observations, we design new probabilistic kernel functions to solve Problems 1.1 and 1.2. The contributions of this paper are summarized as follows.

(1) The proposed kernel functions $K_S^1$ and $K_S^2$ (Eq. (2) and Eq. (3)) rely on a reference-angle-based probabilistic relationship between angles in high-dimensional spaces and projected values. Compared with Eq. (1), the new relationship (Relationship (4)) is deterministic without dependence on asymptotic condition. By theoretical analysis, we show that the proposed kernel functions are effective solvers for the Problems 1.1 and 1.2 (see Lemmas 4.2, 4.3 and 5.1).

(2) By Lemmas 4.2, 4.3, we find that, the smaller the reference angle is, the more accurate the kernel functions are. To minimize the reference angle, we study the structure of the configuration of projection vectors (Sec. 5). We propose two structures (Alg. 1 and Alg. 2) that perform better than purely random projection (Alg. 3 in Appendix). We establish the relationship between the reference angle and the proposed structures (Lemma B.1 and Fig. 5 in Appendix).

(3) Based on $K_S^1$, we propose a random-projection technique KS1 which can be used for CEOs-based tasks (Pham, 2021; Pham & Liu, 2022; Xu & Pham, 2024). Based on $K_S^2$, we introduce a new routing test called the KS2 test which can be used to accelerate the graph-based Approximate Nearest Neighbor Search (ANNS) (Lu et al., 2024) (Sec. 6).

(4) We experimentally show that KS1 provides a slight accuracy improvement (up to 0.8%) over CEOs. For ANNS, we show that HNSW+KS2 improves the query-per-second (QPS) by $2.5\times - 3\times$ over HNSW and by $10\% - 30\%$ over the state-of-the-art approach HNSW+PEOs (Lu et al., 2024), along with a 5% reduction in index size.

## 2 RELATED WORK

Due to space limitations, we focus on random projection techniques that are closely related to this work. An illustration comparing the proposed random projection technique with others can be found in Sec. C.2 and Fig. 4 in Appendix. Since the proposed kernel function is also used in similarity graphs for ANNS, a comprehensive discussion of ANNS solutions is provided in Appendix C.3.

In high-dimensional Euclidean spaces, the estimation of angles via random-projection techniques, especially Locality Sensitive Hashing (LSH) (Indyk & Motwani, 1998; Andoni & Indyk, 2005; 2008), has a relatively long history. A classical LSH technique is SimHash (Charikar, 2002), whose basic idea is to generate multiple hyperplanes and partition the original space into many cells such that two vectors falling into the same cell are likely to have a small angle between them. Andoni et al. (2015) proposed a different LSH method called Falconn for angular distance, whose basic idea is to find the closest or furthest projection vector to the data vector and record this projection vector as a hash value, leading to better search performance than SimHash. Later, Pham (2021) employed Concomitants of Extreme Order Statistics (CEOs) to identify the projection with the largest or smallest inner product with the data vector, as shown in Lemma 1.3, and recorded the corresponding maximum or minimum projected value to obtain a more accurate estimation than using a hash value alone (Pham & Liu, 2022).

Due to its ease of implementation, CEOs has been employed in several similarity search tasks (Pham, 2021; Andoni et al., 2015; Xu & Pham, 2024), as mentioned in Sec. 1. Additionally, CEOs has been used to accelerate similarity graphs, which are among the leading structures for Approximate Nearest Neighbor Search (ANNS). By swapping the roles of query and data vectors in CEOs, Lu et al. (2024) introduced a space-partitioning technique and proposed the PEOs test, which can be used to compare the objective angle with a fixed threshold under probabilistic guarantees. This test was incorporated into the routing mechanisms of similarity graphs and achieved significant search performance improvements over original graph structures like HNSW (Malkov & Yashunin, 2020) and NSSG (Fu et al., 2022).

## 3 TWO PROBABILISTIC KERNEL FUNCTIONS

In Sec. 3, we aim to propose probabilistic kernel functions for Problems 1.1 and 1.2. First, we introduce some notation. Frequently used symbols in this paper are listed in Table 2 in Appendix. Let $\mathbb{R}^d$ be the ambient vector space. Define $H \in SO(d) \subset \mathbb{R}^{d \times d}$ as a random rotation matrix, where $SO(d)$ denotes the special orthogonal group of $d \times d$ rotation matrices[1]. Let $S = [\boldsymbol{u_1}, \boldsymbol{u_2}, \ldots, \boldsymbol{u_m}] \in \mathbb{R}^{d \times m}$ be an arbitrary fixed set of $m$ points on the unit sphere $\mathbb{S}^{d-1}$. For any vector $\boldsymbol{v} \in \mathbb{S}^{d-1}$, define the reference vector of $\boldsymbol{v}$ with respect to $S$ as $Z_S(\boldsymbol{v}) = \operatorname{argmax}_{\boldsymbol{u} \in S} \langle \boldsymbol{u}, \boldsymbol{v} \rangle$. Let $A_S(\boldsymbol{v})$ denote the inner product with the reference vector with respect to $\boldsymbol{v}$, that is, $A_S(\boldsymbol{v}) = \langle \boldsymbol{v}, Z_S(\boldsymbol{v}) \rangle$. Let $HS = \{Hu : u \in S\}$ denote the rotated configuration of $S$. Next, we introduce two probabilistic kernel functions $K_S^1(\cdot, \cdot)$ and $K_S^2(\cdot, \cdot)$ as follows, where $K_S^1(\cdot, \cdot)$ corresponds to Problem 1.1 and $K_S^2(\cdot, \cdot)$ corresponds to Problem 1.2.

$$K_S^1(\boldsymbol{q}, \boldsymbol{v}) = \langle \boldsymbol{v}, Z_{HS}(\boldsymbol{q}) \rangle \qquad \boldsymbol{v}, \boldsymbol{q} \in \mathbb{S}^{d-1}. \tag{2}$$

$$K_S^2(\boldsymbol{q}, \boldsymbol{v}) = \langle H\boldsymbol{q}, Z_S(H\boldsymbol{v}) \rangle / A_S(H\boldsymbol{v}) \qquad \boldsymbol{v}, \boldsymbol{q} \in \mathbb{S}^{d-1}. \tag{3}$$

Remarks. (1) **(Exploitation of reference angle)** In the design of existing projection techniques such as CEOs (Pham, 2021), Falconn (Andoni et al., 2015), Falconn++ (Pham & Liu, 2022), etc., only the reference vector $Z_S(\cdot)$ is utilized. In contrast, our kernel functions defined in Eq. (2) and Eq. (3)

---

[1]Note that the definition here differs from that in Andoni et al. (2015), where the so-called random rotation matrix is actually a matrix with i.i.d. Gaussian entries.

incorporate not only the reference vector $Z_S(\cdot)$ but also the reference angle information $A_S(\cdot)$ (although the reference angle is not explicitly shown in Eq. (2), its influence will become clear in Lemma 4.2). In fact, the reference angle plays a central role, as it is the key factor controlling the precision of angle estimation (see Lemma 4.2).

(2) (**Generalizations of existing works**) These two kernel functions can also be regarded as generalizations of CEOs and PEOs respectively in a certain sense. Specifically, if $S$ is taken as a point set generated via a Gaussian distribution and $Z_S(\boldsymbol{v})$ is replaced by the reference vector having the maximum inner product with the query, then $K_S^1(\boldsymbol{q}, \boldsymbol{v})$ equals the indicator $\boldsymbol{v}^\top \boldsymbol{u}_{\max}$ used in CEOs. Similarly, if we remove the term $A_S(H\boldsymbol{v})$ and take $S$ to be the same space-partitioned structure as that of PEOs, $K_S^2(\boldsymbol{q}, \boldsymbol{v})$ is similar to the indicator of PEOs. In addition, Eq. (3) reduces to the estimator in Gao & Long (2024) when $S$ is taken to be the hypercube with scalar quantization.

(3) (**Configuration of projection vectors**) Although we do not currently require any specific properties of the configuration of $S$, it is clear that the configuration of $S$ impacts both $K_S^1(\boldsymbol{q}, \boldsymbol{v})$ and $K_S^2(\boldsymbol{q}, \boldsymbol{v})$. We will discuss the structure of $S$ in detail in Sec. 5. Notably, we will see that neither the hypercube adopted in Gao & Long (2024) nor the Gaussian distribution adopted in Pham (2021); Lu et al. (2024) provides the optimal configuration for projection vectors.

## 4 ANALYSIS OF PROBABILITY GUARANTEES

In Sec. 4, we show that the proposed probabilistic kernel functions $K_S^1$ and $K_S^2$ satisfy the probability guarantees of Problem 1.1 and Problem 1.2, respectively. Before proceeding, we first provide a definition that will be used to establish a property of $K_S^2$.

**Definition 4.1.** *Let $\phi_1, \phi_2 \in (0, \pi)$ and let $\theta \in (0, \pi)$ be an arbitrary angle threshold. A probabilistic kernel function $K(\boldsymbol{q}, \boldsymbol{v})$ is called **angle-sensitive** when it satisfies the following two conditions:*

*(1) If $\cos\theta \leq \cos\phi_1 = \langle \boldsymbol{q}, \boldsymbol{v} \rangle$, then $\mathbb{P}[K(\boldsymbol{q}, \boldsymbol{v}) \geq \cos\theta] \geq p_1(\phi_1)$,*

*(2) If $\langle \boldsymbol{q}, \boldsymbol{v} \rangle = \cos\phi_2 < \cos\theta$, then $\mathbb{P}[K(\boldsymbol{q}, \boldsymbol{v}) \geq \cos\theta] < p_2(\phi_2)$,*

*where $p_2(\phi_2)$ is a strictly decreasing function in $\phi_2$ and $p_1(\phi_1) > p_2(\phi_2)$ when $\phi_1 < \phi_2$.*

The definition of the angle-sensitive property is analogous to that of the locality-sensitive hashing property. The key difference is that the approximation ratio $c$ used in LSH is not introduced here, as the angle threshold $\theta$ is explicitly defined, and only angles smaller than $\theta$ are considered valid.

We are now ready to present the following two lemmas for $K_S^1$ and $K_S^2$, which demonstrate that they serve as effective solutions to Problems 1.1 and 1.2, respectively.

**Lemma 4.2.** *(1) Let $d \geq 3$ and $(\boldsymbol{q}, \boldsymbol{v})$ be an arbitrary pair of normalized vectors with angle $\phi \in (0, \pi)$. The conditional CDF of $K_S^1(\boldsymbol{q}, \boldsymbol{v})$ can be expressed as follows:*

$$F_{K_S^1(\boldsymbol{q}, \boldsymbol{v})}(x \mid A_S(\boldsymbol{q}) = \cos\psi) = I_t\left(\frac{d-2}{2}, \frac{d-2}{2}\right), \tag{4}$$

*where $\psi \in (0, \pi)$, $t = \frac{1}{2} + \frac{x - \cos\phi\cos\psi}{2\sin\phi\sin\psi}$, $I_t$ denotes the regularized incomplete Beta function and $x \in [\cos(\phi + \psi), \cos(\phi - \psi)]$.*

*(2) Let $\boldsymbol{q}$, $\boldsymbol{v_1}$ and $\boldsymbol{v_2}$ be three normalized vectors on $\mathbb{S}^{d-1}$ such that $\langle \boldsymbol{q}, \boldsymbol{v_1} \rangle > \langle \boldsymbol{q}, \boldsymbol{v_2} \rangle$. The probability $\mathbb{P}[K_S^1(\boldsymbol{q}, \boldsymbol{v_1}) > K_S^1(\boldsymbol{q}, \boldsymbol{v_2}) | A_S(\boldsymbol{q}) = \cos\psi]$ increases as $\psi$ decreases in $(0, \pi)$. In particular, when $\psi \in (0, \pi/2)$, $\mathbb{P}[K_S^1(\boldsymbol{q}, \boldsymbol{v_1}) > K_S^1(\boldsymbol{q}, \boldsymbol{v_2}) | A_S(\boldsymbol{q}) = \cos\psi] > 0.5$, that is, $K_S^1$ satisfies the probability guarantee in Problem 1.1.*

**Lemma 4.3.** *Let $\psi \in (0, \pi/2)$, that is, $A_S(\boldsymbol{v}) = \cos\psi \in (0, 1)$, and $d \geq 3$. Then $K_S^2$ is an angle-sensitive function. Precisely, $K_S^2$ satisfies the probability guarantee in Problem 1.2, where $\epsilon_1 = 0.5$ and $\epsilon_2 = I_{t'}\left(\frac{d-2}{2}, \frac{d-2}{2}\right) < 0.5$, where $t' = \frac{1}{2} - \frac{\cos\theta - \cos\phi}{2\sin\phi\tan\psi}$.*

Remarks. (1) (**Discussion on boundary values**) When $\phi = 0$ or $\phi = \pi$, $K_S^1$ and $K_S^2$ take fixed values rather than being random variables, and when $\psi = 0$ or $\psi = \pi$, the exact value of $\langle \boldsymbol{q}, \boldsymbol{v} \rangle$ can be directly obtained. Therefore, probability analysis in these cases is meaningless. Additionally, in Lemma 4.3, we adopt the following convention: $p_2(\phi) = 0$ if $t' < 0$.

---

**Algorithm 1** Configuration of $S$ via antipodal projections

---

**Input:** $L$ is the level; $d = Ld'$ is the data dimension; $m$ is the number of vectors in each level
**Output:** $S_{\text{sym}}(m, L)$, which is represented by $mL$ sub-vectors with dimension $d'$.

1 **for** $l = 1$ **to** $L$ **do**
2      Generate $m/2$ points i.i.d. on $\mathbb{S}^{d'-1}$, along with their antipodal counterparts
3      Scale the norm of all $m$ points in this iteration to $1/\sqrt{L}$, and collect the vectors after scaling

---

(2) **(Deterministic relationship for angle testing)** Lemma 4.2 establishes a relationship between the objective angle $\phi$ and the value of the function $K_S^1$. Notably, after computing $Z_S(\cdot)$, the value of reference angle $A_S(\cdot)$ can be obtained automatically. Besides, as will be shown in Sec. 5, with a reasonable choice of $S$, the assumption $A_S(\cdot) > 0$ can always be ensured. Hence, Eq. (4) essentially describes a deterministic relationship. In contrast to the asymptotic relationship of CEOs, Eq. (4) provides an exact relationship without additional assumptions.

(3) **(Effectiveness of kernel functions)** The above two lemmas show that, with a reasonable construction of $S$ such that the reference angle is small with a high probability, $K_S^1$ and $K_S^2$ can effectively address the corresponding angle testing problems. The smaller the reference angle is, the more effective $K_S^1$ and $K_S^2$ become.

(4) **(Gaussian distribution is suboptimal)** The fact that a smaller reference angle is favorable justifies the utilization of $Z_S(\cdot)$ and also implies that the Gaussian distribution is not an optimal choice for configuring $S$, since in this case, the selected reference vector with the largest inner product with the query or data vector is not guaranteed to have the smallest reference angle.

## 5    IMPLEMENTATION AND COMPLEXITY ANALYSIS

We discuss how to configure $S$, and then analyze the complexity of $K_S^1$ and $K_S^2$. Based on the discussion in Sec. 4, we observe that small reference angles are preferred. Thus, given $m$, our goal is to construct a set $S$ of $m$ points on $\mathbb{S}^{d-1}$, denoted by $S_m$, such that $A_{S_m}(\cdot)$ is maximized. Due to the effect of the random rotation matrix $H$, the optimal configurations denoted by $\bar{S}_m$ and $S_m^*$, can be obtained either in the sense of expectation or in the sense of the worst case, respectively:

$$\bar{S}_m = \operatorname*{argmax}_{S = \{\boldsymbol{u_1}, \ldots, \boldsymbol{u_m}\} \subset \mathbb{S}^{d-1}} \{\mathbb{E}_{\boldsymbol{v} \in U(\mathbb{S}^{d-1})}[A_S(\boldsymbol{v})]\}, \tag{5}$$

$$S_m^* = \operatorname*{arg\,max}_{S = \{\boldsymbol{u_1}, \ldots, \boldsymbol{u_m}\} \subset \mathbb{S}^{d-1}} \min_{\boldsymbol{v} \in \mathbb{S}^{d-1}} \max_{1 \le i \le m} \langle \boldsymbol{u_i}, \boldsymbol{v} \rangle, \tag{6}$$

where $U(\mathbb{S}^{d-1})$ denotes the uniform distribution on the sphere. By the definitions of $\bar{S}_m$ and $S_m^*$, they correspond to the configurations that achieve the largest expected value and the largest worst-case value of $A_S(\boldsymbol{v})$, respectively. On the other hand, finding the exact solutions for $\bar{S}_m$ and $S_m^*$ is closely related to the best covering problem, which is highly challenging and remains open in the general case. To the best of the authors' knowledge, the optimal configuration $S_m^*$ is only known when $m \le d + 3$ (Borodachov et al., 2019). In light of this, we provide two configurations of $S$: one relies on random antipodal projections (Alg. 1), and the other is built using multiple cross-polytopes (Alg. 2). Each has its own advantages. Alg. 1 enables the estimation of reference angles, while Alg. 2 can empirically produce slightly smaller reference angles and is more efficient for projection computation.

Before proceeding into the details of algorithms, we introduce a quantity $J(S)$ as follows.

$$J(S) = \mathbb{E}_{\boldsymbol{v} \in U(\mathbb{S}^{d-1})}[A_S(\boldsymbol{v})]. \tag{7}$$

By definition, $J(S)$ denotes the expected value of $A_S(\boldsymbol{v})$ w.r.t. $S$ (which equals the cosine of the reference angle when $\boldsymbol{v} \in \mathbb{S}^{d-1}$). This quantity is consistent with our theory, as a random rotation is applied to $\boldsymbol{v}$ or $\boldsymbol{q}$ in Eq. (2) and Eq. (3). Based on the previous discussion, for a fixed $m$, $J(S)$ is maximized when $S = \bar{S}_m$, which is hard to compute. Thus, we take $N$ to be sufficiently large and let $v_{1,N}, \ldots, v_{N,N}$ be vectors drawn independently and uniformly from $U(\mathbb{S}^{d-1})$. Let $\tilde{J}(S, N) = [\sum_{i=1}^N A_S(\boldsymbol{v_{i,N}})]/N$. By the law of large numbers, we can approximate $J(S)$ by $\tilde{J}(S, N)$ when $N$ is sufficiently large.

Now, we are ready to explain Alg. 1 and Alg. 2 as follows.

---

**Algorithm 2** Configuration of $S$ via multiple cross-polytopes

---

**Input:** $L$ is the level; $d = Ld'$ is the data dimension; $m = 2d'a + b$; $R$ is the maximum number of iterations
**Output:** $S_{\text{pol}}(m, L)$, which is represented by $mL$ sub-vectors with dimension $d'$

1 Generate $N$ points randomly and independently on $\mathbb{S}^{d'-1}$, where $N$ is a sufficiently large number
2 **for** $r = 1$ **to** $R$ **do**
3      **for** $t = 1$ **to** $a$ **do**
4          Generate a random rotation matrix $H \in \mathbb{R}^{d' \times d'}$, and rotate $2d'$ axes in $\mathbb{R}^{d'}$ using $H$
5          Collect the $2d'$ vectors of the cross-polytope after rotation
6      **if** $b > 0$ **then**
7          Repeat the above iteration and select $b/2$ antipodal pairs from the rotated cross-polytope
8      For the generated $S \in \mathbb{S}^{d'-1}$, compute $\tilde{J}(S, N)$ and keep the configuration with the largest value, denoted by $S_{\text{max}}$
9 **for** $l = 1$ **to** $L$ **do**
10      Generate a random rotation matrix $H \in \mathbb{R}^{d' \times d'}$ and rotate the configuration $S_{\text{max}}$ using $H$
11      Scale the norm of all $m$ points in this iteration to $1/\sqrt{L}$ and collect the vectors after scaling

---

(1) **(Utilization of antipodal pairs and cross-polytopes)** We use the antipodal pair or the cross-polytope as our building block for the following three reasons. (i) Since all the projection vectors are antipodal pairs, the evaluation time of projection vectors can be halved. (ii) Both two structures can ensure that the assumption $A_S(\boldsymbol{v}) > 0$ holds, such that the condition in Lemma 4.3 is always satisfied. (iii) The result in Borodachov (2022) shows that, for $m = 2d$, under mild conditions, the $2d$ vertices of a cross-polytope can be proven to have the smallest covering radius, that is, the smallest reference angle in the worst case. Although the results in the case $m > 2d$ are unknown, we can rotate the fixed cross-polytope in random directions to generate multiple cross-polytopes until we obtain $m$ vectors, which explains the steps from 3 to 7 in Alg. 2.

(2) **(Selection from random configurations)** We can generate such $m$ points in the above way many times, which forms multiple $S$'s. By the discussion above, we can use $\tilde{J}(S, N)$ to approximately evaluate the performance of $J(S)$, and thus, among the generated $S$'s, we select the configuration $S_{\text{pol}}$ corresponding to the maximal $\tilde{J}(S, N)$. This explains steps 2 and 8 in Alg. 2.

(3) **(Accuracy boosting via multiple levels)** Increasing $m$ can lead to a smaller reference angle. The analysis in Lu et al. (2024) shows that, for certain angle-thresholding problems requiring high accuracy, an exponential increase in $m$, rather than a linear one, can be effective. Therefore, similar to Lu et al. (2024), we use a product-quantization-like technique (Jégou et al., 2011) to partition the original space into $L$ subspaces (levels), which is adopted in Algs. 1 and 2. By concatenating equal-length sub-vectors from these $L$ subspaces, we can virtually generate $m^L$ normalized projected vectors. As will be shown in Lemma B.1, the introduction of $L$ can significantly decrease the reference angle, and $L$ can also control the trade-off between query accuracy and efficiency.

(4) **(Fast projection computation via multi-cross-polytopes)** In Eq. (2) and Eq. (3), we need to compute $HS\boldsymbol{v}$ in the indexing phase. If the projection time is a concern in practice, we can use the Fast Johnson–Lindenstrauss transformation to accelerate this process. Specifically, we use Alg. 2 with $R = 1$. When $L = 1$, the cost of computing $HS\boldsymbol{v}$ is $O(\max(d, m) \log d)$. When $L > 1$, the cost of computing $HS\boldsymbol{v}$ can be reduced to $O(d \log d + mL \log(d/L))$.

By Alg. 1 and Alg. 2, we obtain structures $S_{\text{sym}}(m, L)$ and $S_{\text{pol}}(m, L)$ virtually containing $m^L$ projection vectors. For $S_{\text{sym}}(m, L)$, we can actually establish a deterministic relationship between $J(S_{\text{sym}}(m, L))$ and $(m, L)$ (see Lemma B.1 in Appendix). On the other hand, in practice, $J(S_{\text{pol}}(m, L))$ is often slightly larger than $J(S_{\text{sym}}(m, L))$, as will be shown in Table 1.

With $S_{\text{sym}}(m, L)$ and $S_{\text{pol}}(m, L)$, we are ready to present a complexity analysis of the two proposed functions, showing that they satisfy the complexity requirement in Problem 1.1 and Problem 1.2.

**Lemma 5.1.** *In the indexing phase, for a fixed dataset $\mathcal{D}$ of size $n$, the complexities of $K_S^1$ and $K_S^2$ are $O(nmd)$ and $O(nd \log d + nmd)$, respectively. In the query phase, $K_S^1$ requires $O(md)$- time for random projection, and for each $v \in \mathcal{D}$, it spends $O(1)$-time for computing $K_S^1(q, v)$, while $K_S^2$ spends $O(d \log d + md)$-time for random projection and random rotation, and for each $v \in \mathcal{D}$, it spends $O(L)$-time for computing $K_S^2(q, v)$, where $L \ll d$. Particularly, if $S = S_{pol}(m, L)$ with $R = 1$,*

*the complexities of $K_S^1$ and $K_S^2$ in the indexing phase can be reduced to $O(\max(d, m)n \log d)$ and $O(nd \log d + nmL \log(d/L))$, respectively, where $m$ is greater than $d/L$ by user-specification.*

# 6 APPLICATIONS TO SIMILARITY SEARCH

## 6.1 IMPROVEMENT ON CEOs-BASED TECHNIQUES

As for $K_S^1$, we can use it to improve CEOs, which is used for MIPS and further applied to accelerate LSH-based ANNS (Andoni et al., 2015) and DBSCAN (Xu & Pham, 2024). Since CEOs was originally designed for inner products, we generalize $K_S^1$ to $K_S^{1'}$ as follows to align with CEOs:

$$K_S^{1'}(\boldsymbol{q}, \boldsymbol{v}) = \|\boldsymbol{v}\| \cdot \langle \boldsymbol{v}, Z_{HS}(\boldsymbol{q}) \rangle \qquad \boldsymbol{v} \in \mathbb{R}^d, \boldsymbol{q} \in \mathbb{S}^{d-1}. \tag{8}$$

It is easy to see that, with two minor modifications, that is, replacing $\boldsymbol{v} \in \mathbb{S}^{d-1}$ with $\boldsymbol{v} \in \mathbb{R}^d$, and replacing $x$ with $\|\boldsymbol{v}\|x$ in Eq. (4), Lemma 4.2 still holds. Therefore, $K_S^{1'}$ can be regarded as a reasonable kernel function for inner products. Then, we can apply $K_S^{1'}$ to the algorithm in Pham (2021); Pham & Liu (2022); Xu & Pham (2024). We only need to make the following modification. In these algorithms, the random Gaussian matrix, which denotes the set of projection vectors, can be replaced by $S_{\text{sym}}$ or $S_{\text{pol}}$, with the other parts unchanged. This substitution does not change the complexity of the original algorithms. To distinguish this projection technique based on $K_S^{1'}$ from CEOs, we refer to it as **KS1** (see Alg. 5 for the projection structure of KS1). In the experiments, we will demonstrate that KS1 yields a slight improvement in recall rates over CEOs, owing to a smaller reference angle.

## 6.2 A NEW PROBABILISTIC TEST IN SIMILARITY GRAPH

Lu et al. (2024) proposed probabilistic routing and used it in similarity graphs to accelerate ANNS. Let $dist(\cdot, \cdot)$ be the distance function for ANNS. Each node in the similarity graph represents a data vector. The definition of probabilistic routing is as follows.

**Definition 6.1** (Probabilistic Routing (Lu et al., 2024)). *Given a query vector $\boldsymbol{q}$, a node $v$ in the graph, an error bound $\epsilon$, and a distance threshold $\delta$, for an arbitrary neighbor $w$ of $v$ such that $dist(\boldsymbol{w}, \boldsymbol{q}) < \delta$, if a routing algorithm returns true for $w$ with a probability of at least $1 - \epsilon$, then the algorithm is deemed to be $(\delta, 1 - \epsilon)$-routing.*

Lu et al. (2024) proposed a $(\delta, 1 - \epsilon)$-routing test called PEOs test. Based on $K_S^2$, we propose a new routing test for $\ell_2$ distance, called the **KS2 test**, as follows (see Sec. C.1 and Fig. 3 in Appendix for more details).

$$\sum_{i=1}^{L} \boldsymbol{q_i}^\top \boldsymbol{u}_{\boldsymbol{e}[\boldsymbol{i}]}^{\boldsymbol{i}} \geq A_S(\boldsymbol{e}) \cdot \frac{\|\boldsymbol{w}\|^2/2 - \tau - \boldsymbol{v}^\top \boldsymbol{q}}{\|\boldsymbol{e}\|}. \tag{9}$$

$\boldsymbol{q} \in \mathbb{R}^d$ is the query, $\boldsymbol{v}$ is the visited graph node, $\boldsymbol{w}$ is the neighbor of $\boldsymbol{v}$, and $\boldsymbol{e} = \boldsymbol{w} - \boldsymbol{v}$. $\tau$ is the threshold determined by the result list of graph search. $\boldsymbol{q_i}, \boldsymbol{e_i}$ denote the $i$-th sub-vectors of $\boldsymbol{q}$ and $\boldsymbol{e}$, respectively ($1 \leq i \leq L$). $\boldsymbol{u}_j^i$ denotes the $j$-th sub-vector of the $i$-th subspace of $\boldsymbol{u}$. $\boldsymbol{u}_{\boldsymbol{e}[\boldsymbol{i}]}^{\boldsymbol{i}}$ denotes the reference vector of $\boldsymbol{e_i}$ among all $\{\boldsymbol{u}_j^i\}$'s ($1 \leq j \leq m$). In our experiments, $S$ was set to $S_{\text{sym}}(256, L)$.

During the traversal of the similarity graph, we check the exact distance from graph node $w$ to $q$ only when Ineq. (9) is satisfied; otherwise, we skip the computation of $w$ for efficiency. A complete graph-based algorithm equipped with the KS2 test can be found in Alg. 7. By Lemma 4.3 and the same analysis in Lu et al. (2024), we can easily obtain the following result.

**Corollary 6.2.** *The graph-based search equipped with the KS2 test (9) is a $(\delta, 0.5)$-routing test.*

**Comparison with PEOs.** Since PEOs also uses a Gaussian distribution to generate projection vectors in subspaces, as CEOs does, and does not make use of the reference-angle information, the estimation in Ineq. (9) is more accurate than that of the PEOs test. In addition, the proposed test has two advantages: (1) Ineq. (9) is much simpler than the testing inequality in the PEOs test, resulting in higher evaluation efficiency; (2) Ineq. (9) requires fewer constants to be stored, leading to a smaller index size compared to that of PEOs.

**Complexity analysis.** For the time complexity, for every edge $\boldsymbol{e}$, the computation of the LHS of Ineq. (9) requires $L$ lookups in the table and $L - 1$ additions, while the computation of the RHS of

Table 1: Comparison of recall rates (%) for $k$-MIPS, $k = 10$. The number of projection vectors is 2048. Top-5 projection vectors are probed. Probe@$n$ means top-$n$ points were probed on each probed projection vector. Results are averaged over 10 runs to reduce the bias introduced by random projection.

| Dataset & Method | | Probe@10 | Probe@100 | Probe@1K | Probe@10K |
|---|---|---|---|---|---|
| Word | CEOs(2048) | 34.106 | 71.471 | 90.203 | 98.182 |
| | KS1($S_{\text{sym}}(2048, 1)$) | 34.167 | 71.679 | 90.265 | 98.195 |
| | KS1($S_{\text{pol}}(2048, 1)$) | **34.395** | **72.078** | **90.678** | **98.404** |
| GloVe1M | CEOs(2048) | 1.773 | 6.920 | 24.166 | 63.545 |
| | KS1($S_{\text{sym}}(2048, 1)$) | 1.792 | 7.015 | 24.456 | 64.041 |
| | KS1($S_{\text{pol}}(2048, 1)$) | **1.808** | **7.071** | **24.556** | **64.355** |
| GloVe2M | CEOs(2048) | **2.070** | 6.904 | 21.082 | 54.916 |
| | KS1($S_{\text{sym}}(2048, 1)$) | 2.064 | 6.928 | 21.182 | 55.240 |
| | KS1($S_{\text{pol}}(2048, 1)$) | 1.996 | **6.979** | **21.262** | **55.394** |

Ineq. (9) requires two additions and one multiplication. By using SIMD, we can perform the KS2 test for 16 edges simultaneously. For the space complexity, for each edge, we need to store $L$ bytes to recover $\boldsymbol{q_i}^\top \boldsymbol{u}_{e[i]}^i$, along with two scalars, that is, $A_S(\boldsymbol{e})\|\boldsymbol{w}\|^2/(2\|\boldsymbol{e}\|)$ and $A_S(\boldsymbol{e})/\|\boldsymbol{e}\|$, which are quantized using scalar quantization to enable fast computation of the RHS of Ineq. (9).

## 7 EXPERIMENTS

All experiments were conducted on a PC equipped with an Intel(R) Xeon(R) Gold 6258R CPU @ 2.70GHz. KS1 and KS2 were implemented in C++. The ANNS experiments used 64 threads for indexing and a single CPU thread for searching. We evaluated our methods on six high-dimensional real-world datasets: **Word**, **GloVe1M**, **GloVe2M**, **Tiny**, **GIST**, and **SIFT**. Detailed statistics for these datasets are provided in Appendix D.1. More experimental results can be found in Appendix D.

### 7.1 COMPARISON WITH CEOS

As demonstrated in Sec. 6.1, the results of CEOs are directly used to accelerate other similarity search processes (Pham & Liu, 2022; Xu & Pham, 2024). In this context, we focus solely on the improvement of CEOs itself. We show that KS1, equipped with the structures $S_{\text{sym}}(m, 1)$ and $S_{\text{pol}}(m, 1)$, can slightly outperform CEOs($m$) on the original task of CEOs, that is, $k$-MIPS, where $m$ denotes the number of projection vectors and was set to 2048, following the standard configuration of the original CEOs. Since the only difference among the compared approaches is the configuration of the projection vectors, we use a unified algorithm (see construction of projection structure Alg. 5 and MIPS query processing Alg. 6 in Appendix) with the configuration of projection vectors as an input to compare their recall rates. From the results in Table. 1, we observe that: (1) in most cases, KS1 with the two proposed structures achieves slightly better performance than CEOs, supporting our claim that a smaller reference angle yields a more accurate estimation, and (2) $S_{\text{pol}}$ generally achieves a higher recall rate than $S_{\text{sym}}$, verifying that a configuration closer to the best covering yields better performance. Since the gain of KS1 over CEOs originates from only one difference, that is, the distribution of the generated projected vectors (Gaussian vs. a more diverse distribution on the unit sphere), this result supports our claim that the Gaussian distribution is suboptimal.

### 7.2 ANNS PERFORMANCE

We chose ScaNN (Guo et al., 2020), HNSW (Malkov & Yashunin, 2020), and HNSW+PEOs (Lu et al., 2024) as baselines, where ScaNN is a state-of-the-art quantization-based approach that performs better than IVFPQFS, and HNSW+PEOs (Lu et al., 2024) is a state-of-the-art graph-based approach that outperforms FINGER (Chen et al., 2023) and Glass (Zilliz, 2023). Like HNSW+PEOs, KS2 is implemented on HNSW, dubbed HNSW+KS2. DiskANN (Subramanya et al., 2019) is excluded because it is orthogonal to KS2 and focuses on optimization for external storage. ADSampling (Gao & Long, 2023) is excluded because it is designed for a non-SIMD environment. The parameter settings of all compared approaches and additional experimental results can be found in Appendix D.

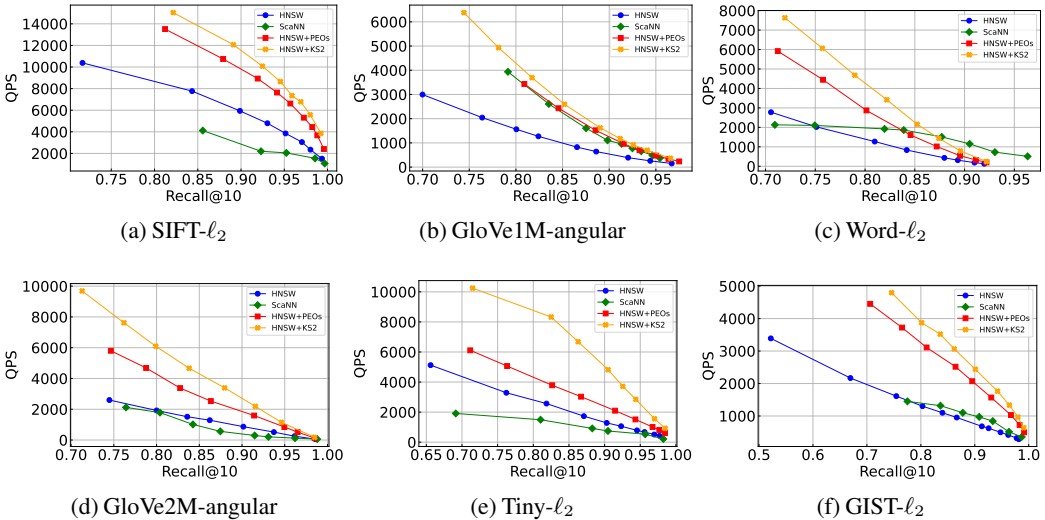

Figure 1: Recall-QPS evaluation of ANNS. $k = 10$.

**(1) Index size and indexing time.** Regarding indexing time, after constructing the HNSW graph, we require an additional 42s, 164s, 165s, 188s, 366s, and 508s to align the edges and build the KS2 testing structure on **Word**, **GloVe1M**, **GIST**, **GloVe2M**, **SIFT**, and **Tiny**, respectively. This overhead is less than 25% of the graph construction time. In practice, users can reduce the parameter efc to shorten indexing time while still preserving the superior search performance of HNSW+KS2. As for the index size, it largely depends on the parameter $L$, which will be discussed later.

**(2) Query performance.** From the results in Fig. 1, we make the following observations. (i) Except for **Word**, HNSW+KS2 achieves the best performance among all compared methods. In particular, HNSW+KS2 accelerates HNSW by a factor of 2.5 to 3, and is 1.1 to 1.3 times faster than HNSW+PEOs, demonstrating the superiority of KS2 over PEOs. (ii) Compared with ScaNN, the advantage of HNSW+KS2 is especially evident in the recall region below 85%, highlighting the effectiveness of the routing test. On the other hand, in the high-recall region for **Word**, ScaNN outperforms HNSW+KS2 due to the connectivity issues of HNSW.

**(3) Impact of $L$.** The only tunable parameter in KS2 is $L$. Generally speaking, the larger $L$ is, the larger the index size is. On the other hand, a larger $L$ can lead to a smaller reference angle and yield better search performance. Hence, $L$ can be used to achieve different trade-offs between index size and search performance. In Fig. 2, we show the impact of $L$ on index size and search performance. From the results, we have the following observations. (i) The index size of HNSW+KS2 is slightly smaller than that of HNSW+PEOs due to the storage of fewer scalars. (ii) When $d' = d/L$ is around 16, HNSW+KS2 achieves the best search performance. This is because a larger $L$ also leads to longer testing time and $d' = 16$ is sufficient to obtain a small enough reference angle.

# 8 CONCLUSIONS

In this paper, we studied two angle-testing problems in high-dimensional Euclidean spaces: angle comparison and angle thresholding. To address these problems, we proposed two probabilistic kernel functions that are based on reference angles and are easy to implement. To minimize the reference angle, we further investigated the structure of the projection vectors and established a relationship between the expected value of the reference angle and the proposed projection vector structure. Based on these two functions, we introduced the KS1 projection and the KS2 test. In the experiments, we showed that KS1 achieves a slight accuracy improvement over CEOs, and that HNSW+KS2 delivers better search performance than the existing state-of-the-art ANNS approaches.

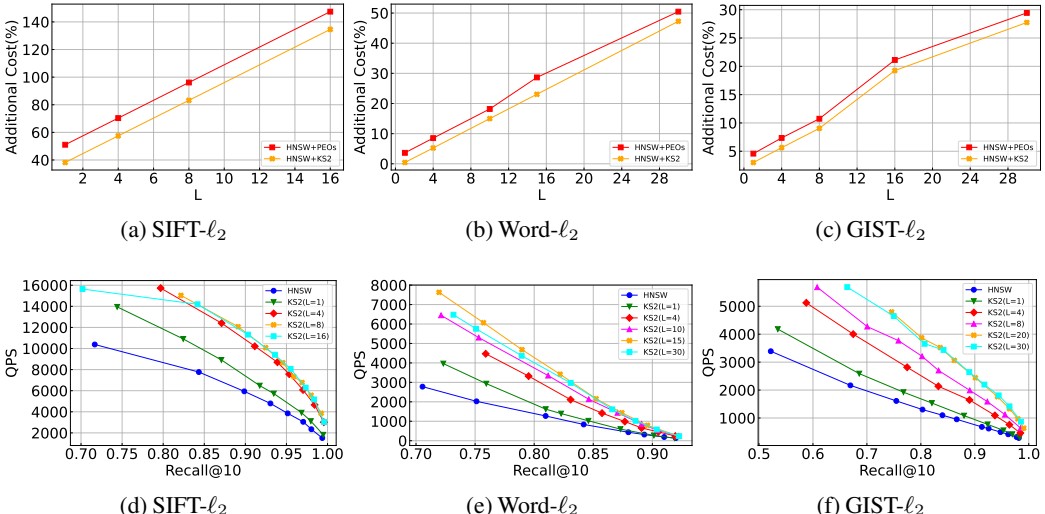

Figure 2: Impact of $L$ (See Appendix D.4 for other datasets). $k = 10$. The y-axis of the upper figures denotes the additional index cost (%) of HNSW+PEOs compared to the original HNSW.

## ACKNOWLEDGEMENTS

This work is supported by JSPS Kakenhi JP22H03594, JP23K17456, JP23K25157, JP23K28096, JP25H01117, JP25K21272, and JST CREST JPMJCR22M2.

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

## A    NOTATIONS

Table 2 lists the main notations used in this paper.

Table 2: Frequently used notations.

| Notation | Explanation |
|---|---|
| $\mathcal{D}$ | Dataset |
| $n$ | Size of Dataset |
| $d$ | Data dimension |
| $\boldsymbol{v}$ | Data vector in $\mathcal{D}$ |
| $\mathbb{S}^{d-1}$ | $d-1$ dimensional sphere |
| $\boldsymbol{q}$ | Query |
| $m$ | The number of projection vectors |
| $\epsilon$ | Error rate |
| $S$ | Fixed set of $m$ points on $\mathbb{S}^{d-1}$ |
| $H$ | Random rotation matrix |
| $Z_S(\boldsymbol{v})$ | $\operatorname{argmax}_{\boldsymbol{u} \in S} \langle \boldsymbol{u}, \boldsymbol{v} \rangle$ |
| $A_S(\boldsymbol{v})$ | $\langle \boldsymbol{v}, Z_S(\boldsymbol{v}) \rangle$ |
| $L$ | The number of space partitions |
| $\boldsymbol{e}$ | Edge in similarity graph |
| $\boldsymbol{w}$ | Neighbor vector of $\boldsymbol{v}$, connected by edge $\boldsymbol{e}$ |
| $\|\boldsymbol{e}\|$ | Length of edge $\boldsymbol{e}$ |
| $\tau$ | Threshold determined by the result list of graph search |
| $\boldsymbol{q_i}, \boldsymbol{e_i}$ | $i$-th sub-vectors of $\boldsymbol{q}$ ($1 \leq i \leq L$), $\boldsymbol{e}$, respectively |
| $\boldsymbol{u}_j^i$ | $j$-th sub-vector of the $i$-th subspace of $\boldsymbol{u}$, where $1 \leq i \leq L$ and $1 \leq j \leq m$ |
| $\boldsymbol{u}_{\boldsymbol{e}[i]}^{\boldsymbol{i}}$ | Reference vector of $\boldsymbol{e_i}$ among all $\{\boldsymbol{u}_j^i\}$'s ($1 \leq j \leq m$) |

## B    PROOF OF LEMMAS

### B.1    PROOF OF LEMMA 4.2

**Proof:** (1) For the first statement, due to the existence of random rotation matrix $H$ and symmetry, we only need to prove the following claim:

**Claim:** Let $\boldsymbol{q}$ and $\boldsymbol{v}$ be two vectors on $\mathbb{S}^{d-1}$ such that the angle between $\boldsymbol{q}$ and $\boldsymbol{v}$ is $\phi$. Let $C$ be a spherical cross-section defined as follows.

$$C = \{\boldsymbol{u} \in \mathbb{S}^{d-1} : \langle \boldsymbol{u}, \boldsymbol{q} \rangle = \cos\psi\}. \tag{10}$$

If $\boldsymbol{u}$ is a vector randomly drawn from $C$, the CDF of $\langle \boldsymbol{v}, \boldsymbol{u} \rangle$ is $I_t(\frac{d-2}{2}, \frac{d-2}{2})$, where $t = \frac{1}{2} + \frac{x - \cos\phi \cos\psi}{2 \sin\phi \sin\psi}$.

**Proof of Claim:** Without loss of generality, we can rotate the coordinate system so that

$$\boldsymbol{q} = (1, 0, \cdots, 0) \in \mathbb{R}^d. \tag{11}$$

Then, by the definition of $\boldsymbol{u}$, $\boldsymbol{u}$ can be written as follows

$$\boldsymbol{u} = (\cos\psi, \sin\psi \cdot \boldsymbol{\omega}) \tag{12}$$

where $\boldsymbol{\omega} \in \mathbb{S}^{d-2} \in \mathbb{R}^{d-1}$ is a unit vector in the subspace orthogonal to $\boldsymbol{q}$. Similarly, $\boldsymbol{v}$ can be expressed as follows.

$$\boldsymbol{v} = (\cos\phi, \sin\phi \cdot \boldsymbol{\eta}) \tag{13}$$

where $\boldsymbol{\eta} \in \mathbb{S}^{d-2}$ is fixed and corresponds to the projection of $\boldsymbol{v}$ onto the orthogonal subspace of $\boldsymbol{q}$. Then $\langle \boldsymbol{v}, \boldsymbol{u} \rangle$ can be written as follows.

$$X := \langle \boldsymbol{v}, \boldsymbol{u} \rangle = \cos\phi \cos\psi + \sin\phi \sin\psi \cdot W \tag{14}$$

where $W = \langle \boldsymbol{\omega}, \boldsymbol{\eta} \rangle$ is a random variable. A well-known fact says that $W$ is related to the Beta distribution as follows.

$$T = \frac{W+1}{2} \sim \text{Beta}\left(\frac{d-2}{2}, \frac{d-2}{2}\right). \tag{15}$$

Therefore, we have

$$F_X(x) = \mathbb{P}(X \le x) = \mathbb{P}(T \le \frac{1}{2} + \frac{x - \cos\phi\cos\psi}{2\sin\phi\sin\psi}). \tag{16}$$

Thus, $F_X(x) = I_t(\frac{d-2}{2}, \frac{d-2}{2})$, where $t = \frac{1}{2} + \frac{x - \cos\phi\cos\psi}{2\sin\phi\sin\psi}$.

(2) For the second statement, due to the existence of random rotation matrix $H$ and symmetry, we only need to prove the following claim.

**Claim:** Given three normalized vectors $v_1$, $v_2$ and $q$, let $\langle q, v_1 \rangle$ be larger than $\langle q, v_2 \rangle$. Let $C$ be the spherical cross-section defined in the proof of statement 1, and let $u$ be a normalized vector drawn randomly from $C$. As $\psi$ decreases, $\mathbb{P}[\langle u, v_1 \rangle > \langle u, v_2 \rangle]$ increases.

**Proof of Claim:** For $u = \cos\psi \cdot q + \sin\psi \cdot \omega'$, where $\omega'$ is a random normalized vector, taking $u$'s inner products with $v_1$ and $v_2$, we obtain:

$$\langle v_1, u \rangle = \cos\psi \langle v_1, q \rangle + \sin\psi \langle v_1, \omega' \rangle. \tag{17}$$

$$\langle v_2, u \rangle = \cos\psi \langle v_2, q \rangle + \sin\psi \langle v_2, \omega' \rangle. \tag{18}$$

Then, we have

$$\mathbb{P}[\langle u, v_1 \rangle > \langle u, v_2 \rangle] \Leftrightarrow \mathbb{P}[\langle v_1, \omega' \rangle - \langle v_2, \omega' \rangle > \Delta_q \cot\psi] \tag{19}$$

where $\Delta_q = \langle v_2, q \rangle - \langle v_1, q \rangle < 0$ and the threshold $\Delta_q \cot\psi$ is naturally set to 0 when $\psi = \pi/2$. As $\psi$ decreases, $\cot\psi$ increases, making $\Delta_q \cot\psi$ smaller. Since $\langle v_1, \omega' \rangle - \langle v_2, \omega' \rangle$ is a random variable due to the existence of $\omega'$, the probability that it exceeds a given threshold increases as the threshold decreases. Thus, as $\psi$ decreases, the probability that the above inequality holds increases.

## B.2 PROOF OF LEMMA 4.3

We consider the following four cases based on the value of $\langle q, v \rangle$.

**Case 1:** $0 < \langle q, v \rangle < 1$.

Let $u$ be a random vector drawn randomly from a spherical cross-section $C'$ defined as follows.

$$C' = \{ u \in \mathbb{S}^{d-1} : \langle u, v \rangle = \cos\psi \}. \tag{20}$$

Next, we construct a simplex with vertices $O, A, B, C$ as follows. We use $\overrightarrow{OA}$ to denote vector $v$, where $O$ denotes the origin. Then we can build a unique hyperplane $H'$ through point $A$ and perpendicular to $v$. We extend $q$ and $u$ along their respective directions to $\overrightarrow{OB}$ and $\overrightarrow{OC}$ such that $B$ and $C$ are on $H'$. Then, we only need to prove the following claim.

**Claim:** Let $O, A, B, C$ be four points in $\mathbb{R}^d$. $\overrightarrow{OA}$ is perpendicular to $\overrightarrow{AB}$, and $\overrightarrow{OA}$ is perpendicular to $\overrightarrow{AC}$. The angle between $\overrightarrow{OA}$ and $\overrightarrow{OC}$ is $\psi$, and the angle between $\overrightarrow{OA}$ and $\overrightarrow{OB}$ is $\phi$. If the angle between $\overrightarrow{AB}$ and $\overrightarrow{AC}$ is $\alpha$, then $\cos\beta$, where $\beta$ denotes the angle between $\overrightarrow{OB}$ and $\overrightarrow{OC}$, can be expressed as follows.

$$\cos\beta = \cos\phi\cos\psi + \sin\phi\sin\psi\cos\alpha. \tag{21}$$

This claim can be easily proved by elementary transformations. Since $\overrightarrow{AB}$ is fixed and $\overrightarrow{AC}$ follows the uniform distribution of a sphere in $\mathbb{R}^{d-1}$, $\cos\alpha$ and $\cos\beta$ are random variables. Therefore, we have the following equalities.

$$\mathbb{P}[K_S^2(q, v) \ge \cos\theta] = \mathbb{P}[\cos\beta \ge \cos\theta\cos\psi]$$
$$= \mathbb{P}[\cos\alpha \ge \frac{\cos\theta - \cos\phi}{\sin\phi\tan\psi}]. \tag{22}$$

We further consider the following two cases.

**Case 1':** $\cos\phi \ge \cos\theta$.

$$\mathbb{P}[K_S^2(q, v) \ge \cos\theta] \ge \mathbb{P}[\cos\alpha \ge 0] = 1/2. \tag{23}$$

**Case 2':** $\cos\phi < \cos\theta$.

---

**Algorithm 3** Configuration of $S$ via random projection

**Input:** $L$ is the level; $d = Ld'$ is the data dimension; $m$ is the number of vectors in each level
**Output:** $S_{\text{ran}}(m, L)$, which is physically represented by $mL$ sub-vectors with dimension $d'$
1 **for** $l = 1$ **to** $L$ **do**
2     Generate $m$ points randomly and independently on $\mathbb{S}^{d'-1}$
3     Scale the norm of all $m$ points in this iteration to $1/\sqrt{L}$ and collect the vectors after scaling

---

By the properties of Beta distribution discussed in the proof of Lemma 4.2 and the property of symmetry of incomplete regularized Beta function, we have

$$\mathbb{P}[K_S^2(\boldsymbol{q}, \boldsymbol{v}) \geq \cos\theta] = I_{t'}\left(\frac{d-2}{2}, \frac{d-2}{2}\right) \tag{24}$$

where $t' = \frac{1}{2} - \frac{\cos\theta - \cos\phi}{2\sin\phi\tan\psi}$. Moreover, since $\tan\psi$ is strictly increasing in $\psi$ when $\psi \in (0, \frac{\pi}{2})$, $\mathbb{P}[K_S^2(\boldsymbol{q}, \boldsymbol{v}) \geq \cos\theta]$ is increasing in $\psi$. On the other hand, it is easy to see that $\mathbb{P}[K_S^2(\boldsymbol{q}, \boldsymbol{v}) \geq \cos\theta]$ is strictly decreasing in $\phi$ when $\phi \in (\theta, \pi)$. Hence, Lemma 4.3 in Case 1 is proved.

**Case 2:** $\langle \boldsymbol{q}, \boldsymbol{v} \rangle = 0$.

In this case, without loss of generality, we can define $\boldsymbol{q}, \boldsymbol{v}, \boldsymbol{u}$ as follows.

$$\boldsymbol{q} = (0, 1, \cdots, 0) \in \mathbb{S}^{d-1}. \tag{25}$$

$$\boldsymbol{v} = (1, 0, \cdots, 0) \in \mathbb{S}^{d-1}. \tag{26}$$

$$\boldsymbol{u} = (\cos\psi, \sin\psi \cdot \boldsymbol{\omega}) \in \mathbb{S}^{d-1} \qquad \boldsymbol{\omega} \sim U(\mathbb{S}^{d-2}). \tag{27}$$

Therefore, we have

$$\mathbb{P}[K_S^2(\boldsymbol{q}, \boldsymbol{v}) \geq \cos\theta] = \mathbb{P}[\cos\alpha \geq \cos\theta/\tan\psi] \tag{28}$$

which is consistent with $\phi = \pi/2$ in Case 1. The following analysis is similar to that in Case 1.

**Case 3:** $-1 < \langle \boldsymbol{q}, \boldsymbol{v} \rangle < 0$.

Instead of $\boldsymbol{v}$ and $\boldsymbol{u}$, we consider $-\boldsymbol{v}$ and $-\boldsymbol{u}$, and construct the simplex based on $\boldsymbol{q}, -\boldsymbol{v}$ and $-\boldsymbol{u}$ as in Case 1. Then, with the reverse of sign, we finally obtain an equation similar to Eq. (21), except with $\alpha$ replaced by $\pi - \alpha$. The following analysis is similar to that of Case 1.

**Case 4:** $\langle \boldsymbol{q}, \boldsymbol{v} \rangle = 1$ or $\langle \boldsymbol{q}, \boldsymbol{v} \rangle = -1$.

In this case, $\langle \boldsymbol{q}, \boldsymbol{u} \rangle = \pm\cos\psi$, and the conclusion is trivial.

### B.3   Estimation of $J(S_{\text{sym}}(m, L))$

As discussed in Sec 5, for $S_{\text{sym}}(m, L)$, we can establish a relationship between $J(S_{\text{sym}}(m, L))$ and $(m, L)$ as follows.

**Lemma B.1.** *Suppose that $d$ is divisible by $L$, and $d = Ld'$, where $d' \geq 3$. Let $c_{d'} = \frac{\Gamma\left(\frac{d'}{2}\right)}{\sqrt{\pi}\,\Gamma\left(\frac{d'-1}{2}\right)}$, $f(y) = c_{d'}(1 - y^2)^{\frac{d'-3}{2}}$ and $F(y) = \int_{-1}^{y} f(t)\,dt$. We have*

$$J(S_{\text{sym}}(m, L)) > m\sqrt{L}\frac{\Gamma(\frac{d+L}{2L})\Gamma(\frac{d}{2})}{\Gamma(\frac{d}{2L})\Gamma(\frac{d+1}{2})}\int_{-1}^{1} yF(y)^{m-1}f(y)\,dy. \tag{29}$$

The RHS of Ineq. (29) actually denotes $J(S_{\text{ran}})$, where $S_{\text{ran}}$ is the configuration of purely random projections (see Alg. 3 for more detail). A numerical computation of the RHS of Ineq. (29) is shown in Fig. 5 in Appendix.

**Proof:** First, we introduce an auxiliary algorithm Alg. 3, which relies on purely random projection. The structure $S$ produced in Alg. 3 is denoted by $S_{\text{ran}}$. We then present the following claim.

**Claim:** $J(S_{\text{sym}}(m, L)) > J(S_{\text{ran}}(m, L))$.

To prove this claim, we introduce the following definition.

**Definition B.2.** *(Stochastic order) Let $X$ and $Y$ be two real-valued random variables. We say that $X <_{st} Y$, if for all $t \in \mathbb{R}$, the CDFs of $X$ and $Y$ satisfy $F_X(t) > F_Y(t)$.*

Let $L = 1$ and fix $m$. We use $X$ to denote the random variable $A_{S_{\mathrm{ran}}}(\boldsymbol{v})$, where $\boldsymbol{v}$ is drawn randomly from $\mathbb{S}^{d-1}$, and $Y$ to denote $A_{S_{\mathrm{sym}}}(\boldsymbol{v})$. Clearly, $X \in [-1, 1]$ and $Y \in [0, 1]$. We have the following:

$$\mathbb{P}(Y > t) > \mathbb{P}(X > t) \quad t \in (0, 1). \tag{30}$$

This can be easily proved, since when the angular radius is less than $\pi/2$, the two spherical caps corresponding to the antipodal pair do not overlap. Therefore, we have $X <_{st} Y$ and $\mathbb{E}[X] < \mathbb{E}[Y]$. This completes the proof of the claim.

Then we only need to focus on $J(S_{\mathrm{ran}}(m, L))$ and prove that it is equal to the RHS of Ineq. (29). Let $\boldsymbol{v} = [\boldsymbol{v_1}, \boldsymbol{v_2}, \cdots, \boldsymbol{v_L}] \in \mathbb{R}^d$ be a vector drawn randomly from $\mathbb{S}^{d-1}$, where $d$ is assumed to be divisible by $L$ and $d = Ld'$. Let $r_L(\boldsymbol{v})$ be the subspace-normalized vector w.r.t. $\boldsymbol{v}$. That is,

$$r_L(\boldsymbol{v}) = [\frac{\boldsymbol{v_1}}{\sqrt{L}\|\boldsymbol{v_1}\|}, \frac{\boldsymbol{v_2}}{\sqrt{L}\|\boldsymbol{v_2}\|}, \dots, \frac{\boldsymbol{v_L}}{\sqrt{L}\|\boldsymbol{v_L}\|}]. \tag{31}$$

Let $T(d, L) = \langle \boldsymbol{v}, r_L(\boldsymbol{v}) \rangle$. For every $i$, let $\boldsymbol{u_i} = \boldsymbol{v_i}/\|\boldsymbol{v_i}\| \in \mathbb{S}^{d'-1}$. Then we select $m$ vectors from $\mathbb{S}^{d'-1}$ randomly and independently. Suppose that, among $m$ generated vectors, $\boldsymbol{w}$ is the vector having the smallest angle to $\boldsymbol{u_i}$, and we use $T'(d', m)$ to denote $\langle \boldsymbol{w}, \boldsymbol{u_i} \rangle$. Since the choice of $\boldsymbol{w}$ for every $\boldsymbol{u_i}$ is independent of the value of $T(d, L)$, we have the following result:

$$\mathbb{E}[A_S(\boldsymbol{v})] = \mathbb{E}[T(d, L)] \times \mathbb{E}[T'(d', m)]. \tag{32}$$

For $\mathbb{E}[T(d, L)]$, since $(\|\boldsymbol{v_1}\|^2, \|\boldsymbol{v_2}\|^2, \dots, \|\boldsymbol{v_L}\|^2)$ follows a Dirichlet distribution with parameters $(\frac{d}{2L}, \frac{d}{2L}, \dots, \frac{d}{2L})$, each $\|\boldsymbol{v_i}\|^2$ marginally follows a Beta distribution with parameters $(\frac{d}{2L}, \frac{d(L-1)}{2L})$. Then by the properties of Beta distribution, we have

$$\mathbb{E}[T(d, L)] = \sqrt{L} \frac{\Gamma(\frac{d+L}{2L})\Gamma(\frac{d}{2})}{\Gamma(\frac{d}{2L})\Gamma(\frac{d+1}{2})}. \tag{33}$$

Next, we consider $\mathbb{E}[T'(d', m)]$. Because of the rotational symmetry of the sphere, the distribution of the inner product $Z = \langle u, v \rangle$ (for fixed $v$ and uniformly random $u$) depends only on the dimension $d'$. It has the following density on $[-1, 1]$:

$$f_Z(z) = c_{d'}(1 - z^2)^{\frac{d'-3}{2}}. \tag{34}$$

Let $Z_1, \dots, Z_m$ be i.i.d. copies of $Z = \langle u, v \rangle$. Then:

$$Y = \max(Z_1, \dots, Z_m). \tag{35}$$

The cumulative distribution function (CDF) of $Z$ is:

$$F_Z(z) = \int_{-1}^{z} f_Z(t)\,\mathrm{d}t. \tag{36}$$

Thus, the CDF of $Y = Y(d', m)$ is:

$$F_Y(y) = \mathbb{P}(Y \le y) = F_Z(y)^m, \tag{37}$$

and the corresponding density is:

$$f_Y(y) = \frac{\mathrm{d}}{\mathrm{d}y} F_Z(y)^m = m F_Z(y)^{m-1} f_Z(y). \tag{38}$$

Therefore, we have the following result.

$$\mathbb{E}[Y] = \int_{-1}^{1} y f_Y(y)\,\mathrm{d}y = m \int_{-1}^{1} y F_Z(y)^{m-1} f_Z(y)\,\mathrm{d}y. \tag{39}$$

Combining the previous results, we get the following result.

$$J(S_{\mathrm{ran}}(m, L)) = m\sqrt{L} \frac{\Gamma(\frac{d+L}{2L})\Gamma(\frac{d}{2})}{\Gamma(\frac{d}{2L})\Gamma(\frac{d+1}{2})} \int_{-1}^{1} y F_Z(y)^{m-1} f_Z(y)\,\mathrm{d}y. \tag{40}$$

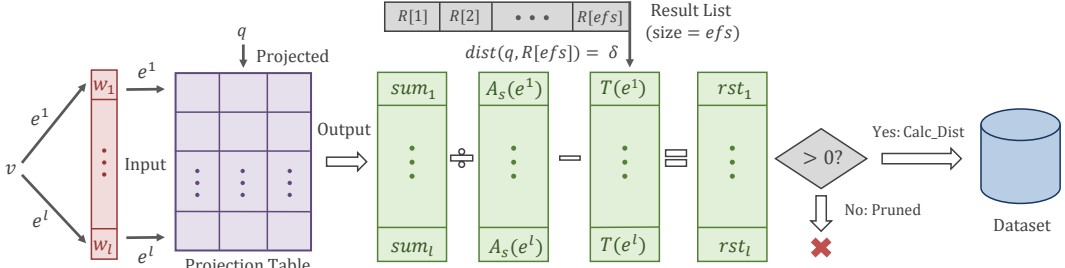

Figure 3: An illustration of the KS2 test.

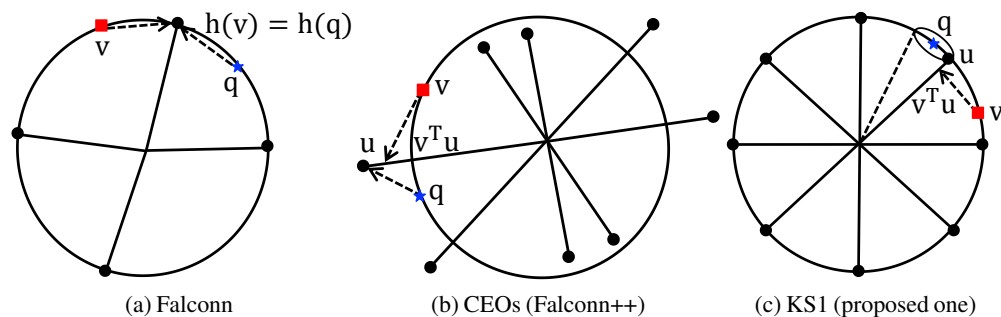

(a) Falconn          (b) CEOs (Falconn++)          (c) KS1 (proposed one)

Figure 4: An illustration of Falconn, CEOs, and the proposed structure KS1.

## C    SUPPLEMENTARY MATERIALS

### C.1    EXPLANATION OF ROUTING TEST IN INEQ. (9)

According to Ineq. (9), there are three core terms I, II, and III, where $I = \sum_{i=1}^{L} \boldsymbol{q_i}^\top \boldsymbol{u}_{\boldsymbol{e[i]}}^{\boldsymbol{i}}$, $II = A_S(\boldsymbol{e})$, and $III = \frac{\|\boldsymbol{w}\|^2/2 - \tau - \boldsymbol{v}^\top \boldsymbol{q}}{\|\boldsymbol{e}\|}$. Here, I/II is exactly the kernel function $K_S^2$; the computation of I depends on PQ (Jégou et al., 2011). III denotes the threshold for $K_S^2$ and is explained in the original paper of PEOs (see the routing test Eq. (9) in Lu et al. (2024)).

### C.2    ILLUSTRATION OF RANDOM PROJECTION TECHNIQUES FOR ANGLE ESTIMATION

An illustration is shown in Fig. 4. In this figure, $\boldsymbol{q}$ represents a query and $\boldsymbol{v}$ is a data vector. For Falconn, if $\boldsymbol{q}$ and $\boldsymbol{v}$ are mapped to the same vector, they are considered close. For CEOs and KS1, the inner product $\boldsymbol{v}^\top \boldsymbol{u}$, where $\boldsymbol{u}$ is the projection vector closest to $\boldsymbol{q}$, is computed to obtain a more accurate estimate of $\boldsymbol{q}^\top \boldsymbol{v}$. The key difference between CEOs and KS1 lies in the structure of the projection vectors. While CEOs uses projection vectors sampled from a Gaussian distribution, KS1 employs a more balanced structure in which all projection vectors lie on the surface of a sphere. By applying a random rotation matrix and constructing a spherical cross-section centered at $\boldsymbol{q}$, we can establish the required statistical relationship between $\boldsymbol{q}^\top \boldsymbol{u}$ and $\boldsymbol{q}^\top \boldsymbol{v}$.

### C.3    ANNS AND SIMILARITY GRAPHS

As one of the most fundamental problems, approximate nearest neighbor search (ANNS) has seen a surge of interest in recent years, leading to the development of numerous approaches across different paradigms. These include tree-based methods (Curtin et al., 2014), hashing-based methods (Andoni & Indyk, 2008; Lei et al., 2019; Andoni & Beaglehole, 2022; Andoni et al., 2015; Pham & Liu, 2022), vector quantization (VQ)-based methods (Jégou et al., 2011; Ge et al., 2014; Babenko & Lempitsky, 2016; Guo et al., 2020), learning-based methods (Gupta et al., 2022; Li et al., 2023), and graph-based methods (Malkov & Yashunin, 2020; Subramanya et al., 2019; Fu et al., 2019; 2022; Gao & Long, 2023; Peng et al., 2023; Xie et al., 2025; Yang et al., 2025).

---

**Algorithm 4** Graph-based ANNS with routing

---

**Input:** Query $q$, # results $K$, graph index $G$
**Output:** $K$-NN of $q$
4  $R \leftarrow \emptyset$ // an ordered list of results, $|R| \leq efs$
5  $P \leftarrow \{$ entry node $v_0 \in G \}$ // a priority queue
6  **while** $P \neq \emptyset$ **do**
7      $\quad v \leftarrow P.pop()$; **foreach** unvisited neighbor $u$ of $v$ **do**
8          $\quad\quad$ **if** $|R| < efs$ **then** $\delta \leftarrow \infty$;
9          $\quad\quad$ **else** $p \leftarrow R[efs], \delta \leftarrow dist(p, q)$;
10         $\quad\quad$ **if** RoutingTest($u, v, q, \delta$) = **true then**
11             $\quad\quad\quad$ **if** $dist(u, q) < \delta$ **then**
12                 $\quad\quad\quad\quad$ $R.push(u); P.push(u)$;

13  **return** ($\{ R[1], \dots, R[K] \}$)

---

Among these, graph-based methods are widely regarded as the state-of-the-art. Currently, three main types of optimizations are used to enhance their search performance: (1) improved edge-selection strategies, (2) more effective routing techniques, and (3) quantization of raw vectors. These approaches are generally orthogonal to one another. Since PEOs belongs to the second category, we briefly introduce several highly relevant works below. TOGG-KMC (Xu et al., 2021) and HCNNG (Muñoz et al., 2019) use KD-trees to determine the direction of the query and restrict the search to points in that direction. While this estimation is computationally efficient, it results in relatively low query accuracy, limiting their improvements over HNSW (Malkov & Yashunin, 2020). FINGER (Chen et al., 2023) examines all neighbors and estimates their distances to the query. For each node, FINGER locally generates promising projection vectors to define a subspace, then uses collision counting which is similar to SimHash to approximate distances within each visited subspace. Learn-to-Route (Baranchuk et al., 2019) learns a routing function using auxiliary representations that guide optimal navigation from the starting vertex to the nearest neighbor. Recently, Lu et al. (2024) proposed PEOs, which leverages space partitioning and random projection techniques to estimate a random variable representing the angle between each neighbor and the query vector. By aggregating projection information from multiple subspaces, PEOs substantially reduces the variance of this estimated distribution, significantly improving query accuracy.

Recently, a related quantization method RaBitQ (Gao & Long, 2024) was proposed and used for the acceleration of similarity graphs (Gou et al., 2025). Notably, RaBitQ can be viewed as a special case of KS2: when $m = 2$ and $L = d$, RaBitQ has exactly the same projection structure as KS2. On the other hand, Lu et al. (2024) shows that when $L$ is very large, an additional error will increase significantly. Thus, if the tradeoff between index size and search performance is taken into account, choosing $L = d$ may not be ideal. Meanwhile, when RaBitQ is applied to similarity graphs, the additional overhead to accommodate the index is very large, consuming about 2–4 times the size of the dataset.

### C.4    ALGORITHMS RELATED TO KS1 AND KS2

Alg. 5 and Alg. 6 are used to compare the probing accuracies of CEOs and KS1, while Alg. 7 presents the graph-based search equipped with the KS2 test. Because Alg. 6 resembles the probabilistic routing mechanism in Lu et al. (2024), we provide further clarification through Alg. 7.

In Alg. 7, the list $R$ serves as an ordered container of at most $efs$ elements ($efs \geq K$). Using a graph index $G = (V, E)$ constructed over the dataset, ANNS results are obtained by traversing this graph, as summarized in Alg. 7. The search starts from an entry node $v_0 \in G$ and maintains $R$, which holds the current best candidates, together with a priority queue $P$ that stores nodes yet to be explored. A neighbor of the current node is inserted into the priority queue whenever it is closer to $q$ than the farthest element in $R$, or whenever $R$ has not reached its capacity. Nodes are repeatedly popped from the priority queue to explore their neighbors, and the procedure terminates once the queue becomes empty. In practice, additional optimizations—such as early stopping or pruning strategies—are commonly adopted to further accelerate the process (Malkov & Yashunin, 2020).

A straightforward implementation would compute the exact distance between $q$ and every encountered neighbor. However, a routing test can be used to decide whether an exact distance evaluation is needed.

---

**Algorithm 5** Construction of the Projection Structure of KS1

---

**Input:** $\mathcal{D}$ is the dataset with cardinality $n$, and $S$ is the configuration of $m$ projection vectors (CEOs: $m/2$ random Gaussian vectors along with their antipodal vectors; KS1: $S_{\text{sym}}(m, 1)$ or $S_{\text{pol}}(m, 1)$)
**Output:** $m$ projection vectors, each of which is associated with a $n$-sequence of data ID's

1  For every $x_i \in \mathcal{D}$ ($1 \leq i \leq n$) and each $u_j \in S$ ($1 \leq j \leq m$), compute $\langle \boldsymbol{x_i}, \boldsymbol{u_j} \rangle$
2  For each $u_j \in S$, sort $x_i$ in descending order of $\langle \boldsymbol{x_i}, \boldsymbol{u_j} \rangle$ and obtain $x_{[1]}^j, \ldots x_{[n]}^j$

---

**Algorithm 6** Query Phase for MIPS

---

**Input:** $q$ is the query; $\mathcal{D}$ is the dataset; $\mathcal{I}$ is the index structure returned by Alg. 5; $k$ denotes the value of top-$k$; $s_0$ is the number of scanned top projection vectors; and #probe denotes the number of probed points for each projection vector
**Output:** Top-$k$ MIP results of $q$

1  Among the $m$ projection vectors, find the top-$s_0$ projection vectors closest to $q$
2  For each projection vector $u_l$ in the top-$s_0$ set ($1 \leq l \leq s_0$), scan the top-#probe points in the sequence associated with $u_l$, and compute the exact inner products of these points with $q$
3  Maintain and return the top-$k$ points among all scanned candidates

---

**Algorithm 7** Graph-based ANNS with the KS2 test

---

**Input**  :$q$ is the query, $k$ denotes the value of top-$k$, $G$ is the similarity graph
**Output** :Top-$k$ ANNS results of $q$

1  $R \leftarrow \emptyset$ ;                                                                    /* an ordered list of results, $|R| \leq efs$ */
2  $P \leftarrow \{$ entry node $v_0 \in G \}$ ;                                                 /* a priority queue */
3  **while** $P \neq \emptyset$ **do**
4  $\quad$ $v \leftarrow P.pop()$
5  $\quad$ **foreach** unvisited neighbor $w$ of $v$ **do**
6  $\quad\quad$ **if** $|R| < efs$ **then** $\delta \leftarrow \infty$;
7  $\quad\quad$ **else** $v' \leftarrow R[efs], \delta \leftarrow dist(\boldsymbol{v'}, \boldsymbol{q})$;
8  $\quad\quad$ **if** KS2_Test($\boldsymbol{w}, \boldsymbol{v}, \boldsymbol{q}, \delta$) = **true** (Ineq. (9)) **then**
9  $\quad\quad\quad$ **if** $dist(\boldsymbol{w}, \boldsymbol{q}) < \delta$ **then**
10 $\quad\quad\quad\quad$ $R.push(\boldsymbol{w}), P.push(\boldsymbol{w})$

11 **return** ($\{ R[1], \ldots, R[k] \}$)

---

The probabilistic routing technique (Lu et al., 2024) is one such example. Meanwhile, the proposed function $K_S^2$ naturally leads to an alternative routing criterion, given by Ineq. (9), which can also be used to determine whether a neighbor should undergo exact distance calculation.

### C.5    NUMERICAL COMPUTATION OF REFERENCE ANGLE

Fig. 5 shows the numerical values of the lower bounds (the RHS of Ineq. (29)) under different pairs of $(m, L)$. From the results, we observe that increasing $L$ significantly raises the cosine of the reference angle, whereas a linear increase in $m$ leads to only a slow growth in the cosine of the reference angle, which explains why we need to introduce parameter $L$ into our projection structure.

## D    ADDITIONAL EXPERIMENTS

### D.1    DATASETS AND PARAMETER SETTINGS

The statistic of six datasets used in this paper is shown in Tab. 3. The parameter settings of all compared methods are shown as follows.

(1) **CEOs:** The number of projection vectors, $m$, was set to 2048, following the standard setting in Pham (2021).

(2) **KS1:** For KS1, $L$ was fixed to 1, as multiple levels are not necessary for angle comparison. The number of projection vectors, $m$, was also set to 2048, consistent with CEOs. We evaluated both KS1($S = S_{\text{sym}}(2048, 1)$) and KS1($S = S_{\text{pol}}(2048, 1)$).

Table 3: Dataset statistics.

| Dataset | Data Size | Query Size | Dimension | Type | Metric |
|---------|-----------|------------|-----------|------|--------|
| Word | 1,000,000 | 1,000 | 300 | Text | $\ell_2$&inner product |
| GloVe1M | 1,183,514 | 10,000 | 200 | Text | angular&inner product |
| GloVe2M | 2,196,017 | 1,000 | 300 | Text | angular&inner product |
| SIFT | 10,000,000 | 1,000 | 128 | Image | $\ell_2$ |
| Tiny | 5,000,000 | 1,000 | 384 | Image | $\ell_2$ |
| GIST | 1,000,000 | 1,000 | 960 | Image | $\ell_2$ |

(3) **HNSW:** $M$ was set to 32. The parameter efc was set to 1000 for **SIFT**, **Tiny**, and **GIST**, and to 2000 for **Word**, **GloVe1M**, and **GloVe2M**.

(4) **ScaNN:** The Dimensions_per_block was set to 4 for **Tiny** and **GIST**, and to 2 for the other datasets. num_leaves was set to 2000. The other user-specified parameters were tuned to achieve the best trade-off curves.

(5) **HNSW+PEOs:** Following the suggestions in Lu et al. (2024), we set $L$ to 8, 10, 15, 15, 16, and 20 for the six real datasets, sorted in ascending order of dimension. Additionally, $\epsilon$ was set to 0.2, and $m = 256$ to ensure that each vector ID could be encoded with a single byte.

(6) **HNSW+KS2:** $S$ was fixed to $S_{\text{sym}}(m, L)$. The only tunable parameter is $L$, as $m$ must be fixed at 256 to ensure that each vector ID is encoded with a single byte. Since the parameter $L$ in KS2 plays a similar role to that in PEOs, we set $L$ to the same value in HNSW+PEOs to eliminate the influence of $L$ in the comparison.

## D.2 KS1 VS. CEOS UNDER DIFFERENT SETTINGS

In Tab. 1, we compared the performance of CEOs and KS1 using the top-5 probed projection vectors. Here, we varied the value of $s_0$ in Alg. 6 from 5 to 2 and 10, and present the corresponding results in Tab. 4 and Tab. 5, respectively. We observe that KS1($S_{\text{pol}}$) still achieves the highest probing accuracy in most cases.

## D.3 ANNS RESULTS UNDER DIFFERENT $k$'S

Fig. 6 and Fig. 7 show the comparison results of ANNS solvers under different values of $k$. When $k = 1$, ScaNN performs very well on **Word**, **GloVe1M**, **GloVe2M**, and **Tiny**, which is partly due to the connectivity issue of HNSW on these datasets. In the other cases, HNSW+KS2 achieves the best performance.

## D.4 THE IMPACT OF $L$ ON THE OTHER DATASETS

Fig. 8 shows the impact of $L$ on **GloVe1M**, **GloVe2M**, and **Tiny**, which is largely consistent with the results in Fig. 2.

## D.5 RESULTS OF NSSG+KS2

We also implement KS2 on another state-of-the-art similarity graph, NSSG (Fu et al., 2022). The parameter settings for NSSG are the same as those in Lu et al. (2024). From the results in Fig. 9, we observe that NSSG+KS2 outperforms NSSG+PEOs on all datasets except for **GIST**, which indicates that the superiority of KS2 is independent of the underlying graph structure.

## STATEMENTS ON THE USE OF LARGE LANGUAGE MODELS

We used LLMs to polish writing only. We are responsible for all the materials presented in this work.

Table 4: Comparison of recall rates (%) for $k$-MIPS, $k = 10$. Top-2 projection vectors are probed.

| Dataset & Method | | Probe@10 | Probe@100 | Probe@1K | Probe@10K |
|---|---|---|---|---|---|
| Word | CEOs(2048) | 17.255 | 46.348 | 68.893 | 84.366 |
| | KS1($S_{\mathrm{sym}}(2048, 1)$) | 17.334 | 46.740 | 69.232 | 84.548 |
| | KS1($S_{\mathrm{pol}}(2048, 1)$) | **17.440** | **46.843** | **69.392** | **85.026** |
| GloVe1M | CEOs(2048) | 0.839 | 3.404 | 12.694 | 38.607 |
| | KS1($S_{\mathrm{sym}}(2048, 1)$) | 0.849 | 3.464 | 12.890 | 39.109 |
| | KS1($S_{\mathrm{pol}}(2048, 1)$) | **0.866** | **3.503** | **12.916** | **39.170** |
| GloVe2M | CEOs(2048) | 0.934 | 3.407 | 11.621 | 34.917 |
| | KS1($S_{\mathrm{sym}}(2048, 1)$) | **0.939** | **3.451** | **11.775** | **35.462** |
| | KS1($S_{\mathrm{pol}}(2048, 1)$) | 0.917 | 3.401 | 11.748 | 35.159 |

Table 5: Comparison of recall rates (%) for $k$-MIPS, $k = 10$. Top-10 projection vectors are probed.

| Dataset & Method | | Probe@10 | Probe@100 | Probe@1K | Probe@10K |
|---|---|---|---|---|---|
| Word | CEOs(2048) | 50.772 | 84.545 | 96.620 | 99.882 |
| | KS1($S_{\mathrm{sym}}(2048, 1)$) | 50.698 | 84.470 | 96.643 | 99.869 |
| | KS1($S_{\mathrm{pol}}(2048, 1)$) | **51.238** | **84.865** | **96.869** | **99.910** |
| GloVe1M | CEOs(2048) | 3.012 | 11.301 | 36.293 | 80.995 |
| | KS1($S_{\mathrm{sym}}(2048, 1)$) | 3.047 | 11.401 | 36.604 | 81.307 |
| | KS1($S_{\mathrm{pol}}(2048, 1)$) | **3.069** | **11.451** | **36.857** | **81.781** |
| GloVe2M | CEOs(2048) | **3.574** | 11.252 | 30.695 | 69.406 |
| | KS1($S_{\mathrm{sym}}(2048, 1)$) | 3.567 | 11.256 | 30.741 | 69.668 |
| | KS1($S_{\mathrm{pol}}(2048, 1)$) | 3.515 | **11.368** | **30.940** | **70.016** |

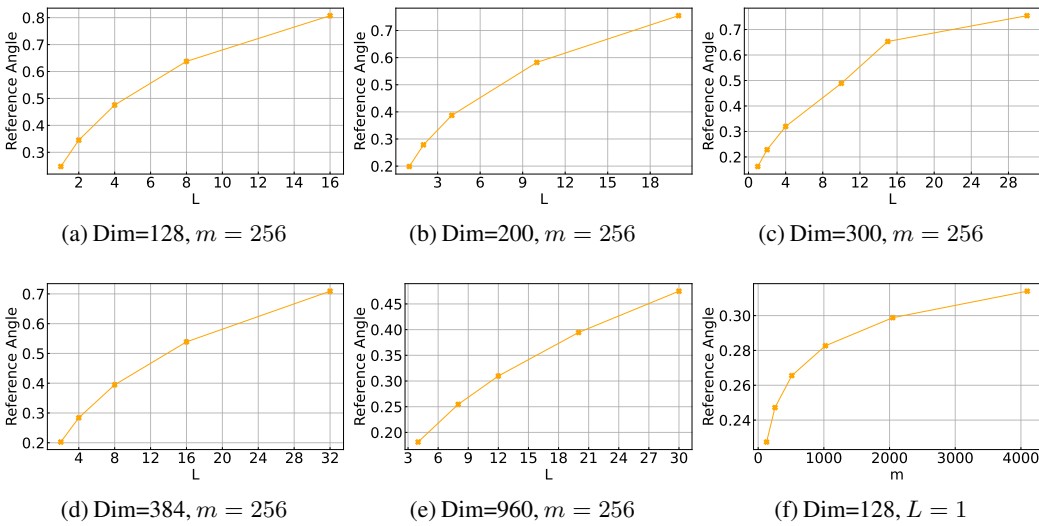

(a) Dim=128, $m = 256$    (b) Dim=200, $m = 256$    (c) Dim=300, $m = 256$

(d) Dim=384, $m = 256$    (e) Dim=960, $m = 256$    (f) Dim=128, $L = 1$

Figure 5: Numerical computation under different $m$'s and $d$'s. The y-axis denotes the cosine of reference angle.

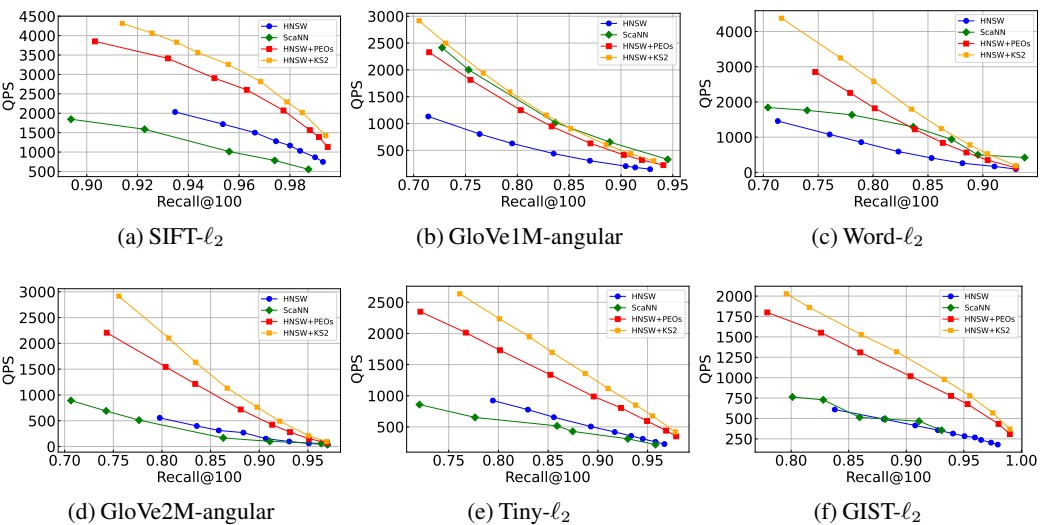

Figure 6: Recall-QPS evaluation of ANNS. $k = 100$.

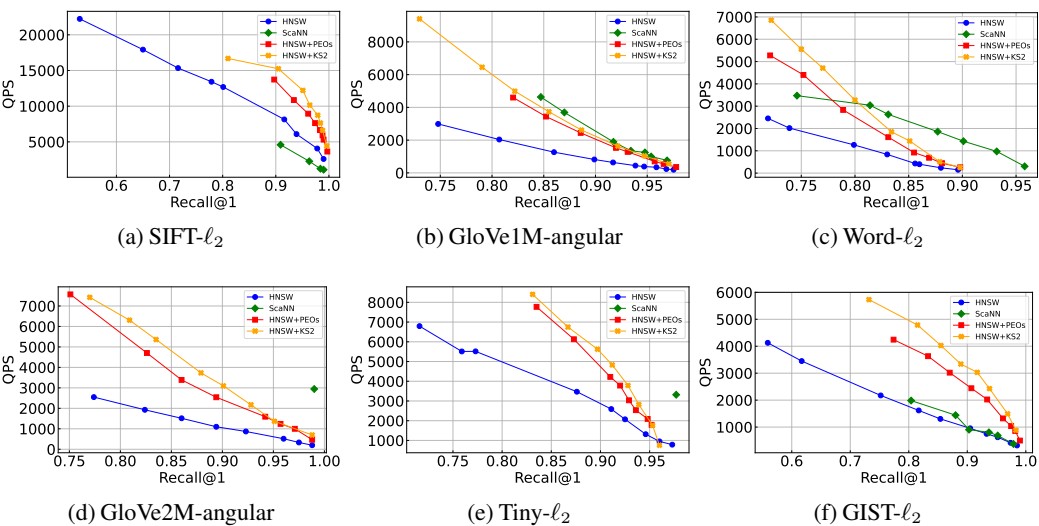

Figure 7: Recall-QPS evaluation of ANNS. $k = 1$.

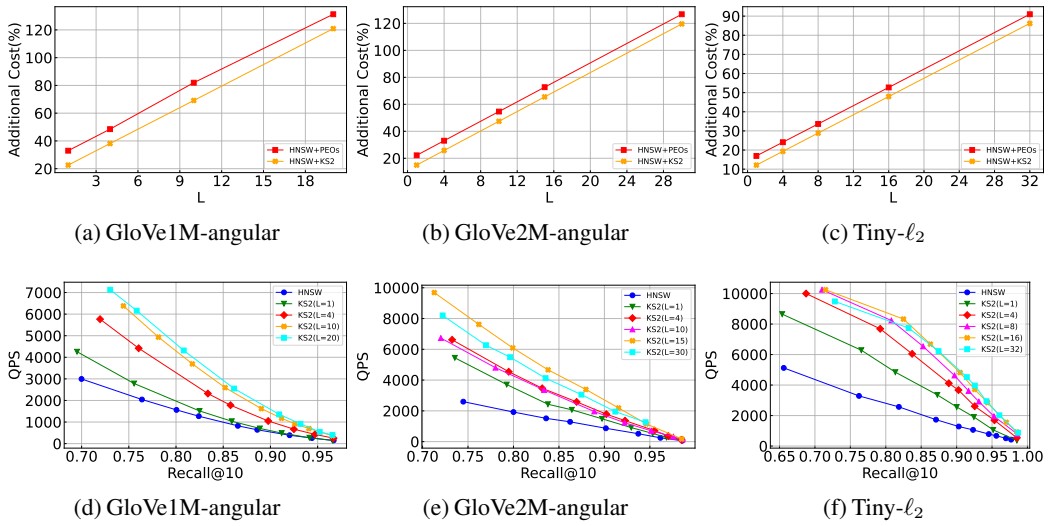

Figure 8: Impact of $L$ on index sizes and search performance. $k = 10$.

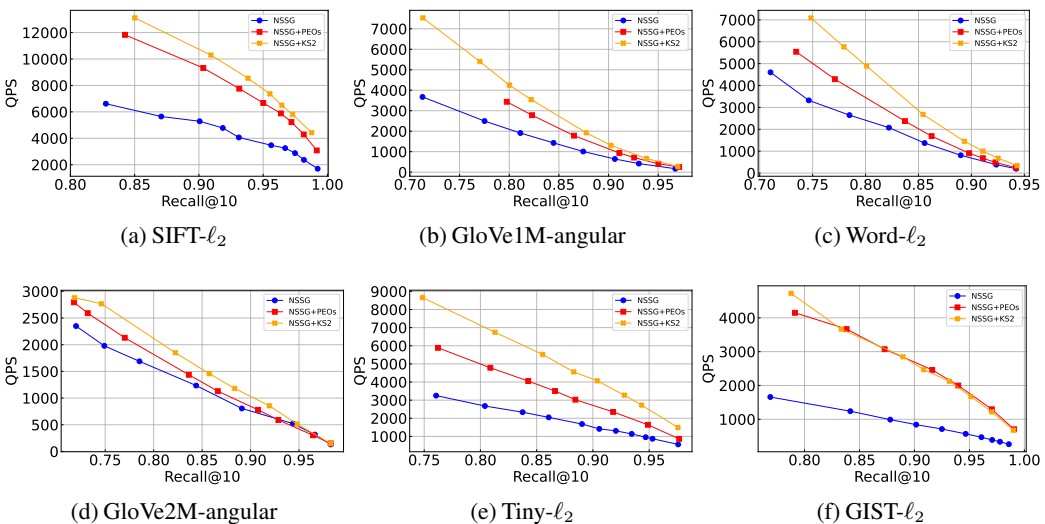

Figure 9: Recall-QPS evaluation of ANNS, with NSSG+KS2. $k = 10$.

