# OpenReview forum: "Probabilistic Kernel Function for Fast Angle Testing"
_ICLR.cc/2026/Conference — ICLR 2026 Oral_

### Official Review · Reviewer_mXUD · 2025-10-29

**Soundness:** 3
**Presentation:** 4
**Contribution:** 4
**Rating:** 8
**Confidence:** 3

**Summary:**

This paper tackles the angle testing problem as part of similarity search in high-dimensional Euclidean spaces. The key idea is to depart from conventional random projection methods based on Gaussian distributions, and instead propose probabilistic kernel functions with deterministic structures that leverage reference angles. Unlike prior approaches, the proposed method does not rely on asymptotic assumptions and theoretically and experimentally outperforms Gaussian-based methods. Applied to approximate nearest neighbor search (ANNS), the proposed approach achieves 2.5–3× higher query throughput (QPS) compared to HNSW.

**Strengths:**

This paper's strengths are as follows.

(1) The paper presents a theoretically rigorous approach to the angle testing problem without asymptotic assumptions, proposing probabilistic kernel functions with deterministic structures.

(2) By designing appropriate projection vector structures, the method improves estimation accuracy and further demonstrates that Gaussian structures are suboptimal.

(3) The proposed method can be easily integrated into existing algorithms such as HNSW, suggesting strong practical applicability.

(4) Extensive experiments on multiple datasets show that the proposed method achieves both high speed and high accuracy.

**Weaknesses:**

This paper's weaknesses are as follows.

(1) Constructing the projection vector structures is computationally expensive and could become a bottleneck for extremely large-scale datasets.

(2) Minor typographical issue found: Line 153: “ZS(·))” → “ZS(·)”.

**Questions:**

My questions about this paper are as follows.

(1) Since the proposed method is based on probabilistic kernel functions, it presumably involves stochastic errors. Is it possible to predict or estimate the error rate analytically?

(2) How does the error rate depend on the projection vector structure? Can this relationship be analyzed theoretically?

(3) How should the number of projection vectors m and the number of subspaces L be optimized? These parameters likely depend on the data distribution, but what practical optimization strategy do the authors envision for real-world applications?

---

> ### Author Response · Authors · 2025-11-20
>
> We thank the reviewer for acknowledging our contributions and giving insightful comments and suggestions. Our responses to the reviewer's questions are as follows.
>
> > (Weakness 1) Constructing the projection vector structures is computationally expensive and could become a bottleneck for extremely large-scale datasets.
>
> We acknowledge this weakness. The additional index size is difficult to reduce. Nonetheless, the index size of our method is smaller than that of PEOs, as shown in Fig. 2 (a-c). Moreover, users can adjust the parameter $L$ to balance the index size and search efficiency.
>
> As for indexing time, we reported the indexing times of KS2 on line 417 of the paper. The total indexing time of KS2 is much smaller than the construction time of HNSW.
>
> In addition, as an extension of this work, we are now exploring in the direction of applying KS2 to accelerate graph construction, so that the indexing time of HNSW+KS2 can be smaller than that of standard HNSW.
>
> > (Question 1) Since the proposed method is based on probabilistic kernel functions, it presumably involves stochastic errors. Is it possible to predict or estimate the error rate analytically?
>
> In Lemma 4.3, we show the error bounds analytically. Roughly speaking, the positive nodes are computed with probability at least $\epsilon_1 = 0.5$, while the false nodes are computed with probability $\epsilon_2 < 0.5$, where $\epsilon_2$ is given in Lemma 4.3 and its value depends on the gap to the angle threshold.
>
> > (Question 2) How does the error rate depend on the projection vector structure? Can this relationship be analyzed theoretically?
>
> The larger the number of projection vectors is, the smaller the reference angle becomes, and thus the more accurate the estimation is. The former statement is trivial, and the latter is shown in Eq. (4) in Lemma 4.2, which illustrates the relationship between the reference angle and the estimation accuracy.
>
> > (Question 3) How should the number of projection vectors m and the number of subspaces L be optimized? These parameters likely depend on the data distribution, but what practical optimization strategy do the authors envision for real-world applications?
>
> $m$ is set to 256, so that every projection vector in a single subspace can be represented using exactly one byte.
>
> For the choice of $L$, we follow the setting in the PEOs paper (Lu et al., 2024), where the impact of $L$ is theoretically analyzed, and the optimal value depends only on the dimension rather than the underlying data distribution. We conducted experiments (Fig. 2 a-c) to show how different values of $L$ affect the tradeoff between index size and search performance. Our experimental results show that before reaching the optimal value, a larger $L$ leads to better performance but also a larger index size. Therefore, we suggest that users adjust $L$ based on the memory availability.

---

> > ### Comment · Reviewer_mXUD · 2025-11-26
> >
> > Thank you for the detailed explanation. Even considering the concerns regarding computational cost, my evaluation remains positive. I hope that this issue can be addressed in future work.

---

> > > ### Author Response · Authors · 2025-11-26
> > > **Re: Official Comment by Reviewer mXUD**
> > >
> > > Thanks for the recognition of our contribution. We acknowledge the indexing cost as a limitation of the proposed method, and reducing the indexing time is the goal of our ongoing work.

---

### Official Review · Reviewer_yHFc · 2025-11-01

**Soundness:** 4
**Presentation:** 3
**Contribution:** 3
**Rating:** 8
**Confidence:** 3

**Summary:**

The most commonly used metrics in approximate nearest neighbor, i.e. Euclidean distance, cosine similarity, and inner product, can crucially be reduced to just inner product computation with appropriate preprocessing. The paper proposes two probabilistic kernel functions that approximate (1) the comparison of two inner products $\langle q, v_1\rangle$ and $\langle q, v_2\rangle$ and (2) whether $\langle q, v\rangle$ exceeds a given threshold. These approximations are much faster than the corresponding computations, making it possible to significantly accelerate approximate nearest neighbor search algorithms.

The proposed kernel functions are based on reference angles. Based on these functions, the authors introduce the KS1 test for CEOs tasks such as maximum inner product search and the KS2 test as a routing test in graph-based approximate nearest neighbor search. In the authors' experiments, the KS1 test yields slight improvements, while the KS2 test applied to the HNSW algorithm yields significant improvements.

**Strengths:**

The paper proposes an interesting approach, angle testing, which has a central role in approximate nearest neighbor search. The authors propose methods for both angle comparison and thresholding. The provided theory is neat and well-motivated.

The proposed KS2 test can be generally applied to many different graph-based approximate nearest neighbor methods, and is amenable to an efficient SIMD implementation that yields a significant improvement in throughput when combined with the popular HNSW algorithm. This combination is also more efficient than combining HNSW with the earlier PEOs approach.

The experiments provided by the authors are comprehensive and use standard benchmark datasets.

**Weaknesses:**

In practice, the improvement provided by KS1 over CEOs is very minor. The improvement of HNSW+KS2 over the earlier HNSW+PEOs is slightly larger but still relatively small.

The state-of-the-art methods for approximate nearest neighbor search combine graphs and quantization, e.g. Glass combines graphs with scalar quantization and SymphonyQG [1] combines graphs with RaBitQ, yet these are not included in the comparisons. The authors mention that they do not compare to e.g. Glass as it was deemed less efficient than PEOs in the corresponding paper, but experimental results in the PEOs paper do not seem to align with e.g. the results of ANN-benchmarks [2]. The KS2 test is not applicable to quantized vectors in e.g. uint8 or binary precision which are increasingly popular due to the high dimensionality of modern embedding datasets.

A weakness in presentation is that the numerous references to analysis and explanations in the PEOs paper make it difficult to understand the paper without a thorough reading of the PEOs paper.

[1] Gou et al. SymphonyQG: Towards Symphonious Integration of Quantization and Graph for Approximate Nearest Neighbor Search. Proceedings of the ACM on Management of Data. 2025.

[2] Aumüller et al. ANN-benchmarks: A benchmarking tool for approximate nearest neighbor algorithms. Information Systems, 2020.

**Questions:**

- Could the authors point out where in the provided code release the SIMD implementation of the KS2 test is located? I tried briefly looking in the code but was unable to figure out where in the code it was.

- The integration of KS2 into HNSW moderately increases the index size and indexing time. How do these compare to HNSW+PEOs?

- Have the authors considered testing HNSW+KS2 by integrating it to a standard benchmark such as ANN-benchmarks such that it is easier to reliably compare results?

---

> ### Author Response · Authors · 2025-11-20
>
> We thank the reviewer for acknowledging our contributions and giving insightful comments and suggestions. Our responses to the reviewer's questions are as follows.
>
> > (Weakness 1) The state-of-the-art methods for approximate nearest neighbor search combine graphs and quantization, e.g. Glass combines graphs with scalar quantization and SymphonyQG [1] combines graphs with RaBitQ, yet these are not included in the comparisons. The authors mention that they do not compare to e.g. Glass as it was deemed less efficient than PEOs in the corresponding paper, but experimental results in the PEOs paper do not seem to align with e.g. the results of ANN-benchmarks [2].
>
>
> We conducted an experiment comparing Glass with HNSW on the GIST dataset (k = 10), which is also used in the ANN-benchmarks. We strictly followed the parameter settings used in the ANN-benchmarks: R = 32 and L = 200. The Recall–QPS results are reported below, showing that Glass is  faster than HNSW but slower than PEOs, consistent with the observations reported in the PEOs paper.
>
> | Method | QPS@recall=0.8 | QPS@recall=0.9 | QPS@recall=0.95 |
> |:-------|-----------:|-----------:|------------:|
> | HNSW   |      1301 |       770 |        476 |
> | Glass  |      1648 |       826 |        455 |
> | PEOs   |      3221 |      2051 |       1129 |
> | KS2    |   **3910** |   **2425** |   **1443** |
>
>
> As for the discrepancy from the results reported in the ANN-benchmarks, we suspect that it arises from the difference in hardware. Note that some ANNS methods rely on advanced hardware architectures for acceleration. Although we are not entirely certain of the cause, we plan to submit our method to the ANN-benchmarks so that it can be evaluated and compared on their hardware.
>
> SymphonyQG is essentially NSG + RaBitQ, where NSG (Fu et al., 2019) is a type of similarity graph, along with some techniques to avoid random access, which are applicable to KS2 without difficulty. Therefore, we focus on the comparison between RaBitQ and KS2, which can be summarized as follows.
>
> (1) RaBitQ can be viewed as a special case of KS2: When $m = 2$ and $L = d$, RaBitQ has exactly the same projection structure as KS2. On the other hand, the PEOs paper (Lu et al., 2024) shows that when $L$ is very large, an additional error will significantly increase. Thus, if the tradeoff between index size and search performance is considered, $L = d$ might not be a good choice.
>
> (2) In our theoretical analysis of KS2, our relationship is strict (see Eq. 4), while the error bound of RaBitQ is not strict. This is because our analysis is built upon the conditional probability of the error, whereas the error is embedded in the estimator of RaBitQ.
>
> (3) When RaBitQ is applied to similarity graphs, the additional overhead to accommodate the index is very large, about 2X-3X larger than standard HNSW on high-dimensional datasets such as GIST, due to $L = d$. On the other hand, the index size of HNSW + KS2 is only about 0.2X larger than HNSW under a moderate $L$ since $L \ll d$ (see Fig. 2). Furthermore, KS2 takes the index size into account and allows users to adjust $L$ to control the tradeoff between index size and search performance (see Fig. 2).
>
> We will add the above discussion in the revised paper, so that users can better understand the positioning of KS2 compared to RaBitQ.
>
> > (Weakness 2) A weakness in presentation is that the numerous references to analysis and explanations in the PEOs paper make it difficult to understand the paper without a thorough reading of the PEOs paper.
>
> We will add a detailed explanation in the appendix to introduce PEOs.

---

> > ### Author Response · Authors · 2025-11-20
> >
> > > (Question 1) Could the authors point out where in the provided code release the SIMD implementation of the KS2 test is located? I tried briefly looking in the code but was unable to figure out where in the code it was.
> >
> > The SIMD implementation is in ```KS2/l2/hnswalg.h```, lines 745–793, which corresponds to the KS2 test in Eq. (9).
> >
> > In this part, we store several scalars that can be precomputed to accelerate the evaluation of Eq. (9). The difference between the right-hand side of Eq. (9) and the left-hand side is stored in the array Thres1. If ```Thres1[i] <= 0``` on line 797 holds, then the i-th neighbor node passes the KS2 test, and we further compute its distance to the query.
> >
> > We will clean up the source code to enhance the readability.
> >
> > > (Question 2) The integration of KS2 into HNSW moderately increases the index size and indexing time. How do these compare to HNSW+PEOs?
> >
> > Our test function requires fewer scalars to be stored, and therefore the index size is smaller than that of PEOs. Please refer to Fig. 2 (a-c) for the comparisons between KS2 and PEOs in terms of index size.
> >
> > For the indexing time without considering the impact of the underlying similarity graph, KS2 is larger than PEOs. We reported the indexing times of KS2 on line 417 of the paper. The total indexing time of KS2 is much smaller than the construction time of HNSW.
> >
> > In addition, as an extension of this work, we are now exploring in the direction of applying KS2 to accelerate graph construction, so that the indexing time of HNSW+KS2 can be smaller than that of standard HNSW.
> >
> > > (Question 3) Have the authors considered testing HNSW+KS2 by integrating it to a standard benchmark such as ANN-benchmarks such that it is easier to reliably compare results?
> >
> > We are cleaning the source code and evaluating the aforementioned fast indexing of HNSW+KS2. We will submit our implementation to the ANNS benchmarks within the next 2-3 months.

---

> > > ### Comment · Reviewer_yHFc · 2025-11-24
> > >
> > > Thank you for your detailed response.
> > >
> > > I appreciate that you have also provided experimental results comparing your method to Glass. One thing to be aware of though is that to my understanding the hyperparameters in ANN-benchmarks are in actuality tuned for $k = 100$. E.g. in https://vector-index-bench.github.io/tradeoff.html at $k = 100$ one finds that Glass and SymphonyQG are typically 1.5-2.5x faster than HNSW. Though indeed there can also be a lot of variation in experimental results depending on hardware etc. and I'm satisfied with the provided results.
> > >
> > > I trust that the authors will include the additional commentary on RaBitQ and an explanation of PEOs in the final revision of the paper and therefore I will retain my positive assessment of the paper.

---

> > > > ### Author Response · Authors · 2025-11-24
> > > > **Re: Official Comment by Reviewer yHFc**
> > > >
> > > > Thanks for your positive feedback. We make sure that the commentary on RaBitQ and the explanation of PEOs will be included in the next version.

---

### Official Review · Reviewer_XabQ · 2025-11-02

**Soundness:** 3
**Presentation:** 2
**Contribution:** 3
**Rating:** 8
**Confidence:** 3

**Summary:**

This paper proposes two projection-based probabilistic kernel functions for fast angle testing in the context of similarity search in high-dimensional Euclidean spaces. Its core observation is that the reference angle can determine the estimation accuracy of angle comparison and testing, which in turn can be determined by the structure of the projection vectors. A detail theoretical analysis is presented to support the effectiveness of the proposed probabilistic kernel functions. Then, algorithms are proposed to compute the projection vectors, together with an analysis of the computational complexity. The proposed probabilistic kernel functions are integrated into high-dimensional similarity search algorithms (Maximum Inner Product Search and Graph-based ANN Search) to enhance their effectiveness and efficiency, which are verified with experiments on six commonly used benchmark vector datasets.

**Strengths:**

1. This paper studies an important problem - high-dimensional similarity search.

2. Detail theoretical analysis is presented to show the effectiveness and correctness of the proposed probabilistic kernel functions.

3. Experimental results are presented to show the empirical effectiveness of the proposed probabilistic kernel functions and algorithms.

4. Source code has been released.

**Weaknesses:**

The empirical performance of the proposed kernel functions is not as strong. It produces marginally higher recall than the CEOs technique (Pham, 2021) as shown in Table 1, while HNSW+KS2 is only 1.1 to 1.3 times faster than HNSW+PEOs. Also, why are Tiny, GIST, and SIFT omitted from Table 1?

It would be good to discuss if the proposed kernel functions are guaranteed to lead to more accurate (and/or more efficient) similarity search than Gaussian-distribution-based kernel functions given the same number of projection vectors. If this is not guaranteed, it would be good to tune down the claim: "can be both theoretically and experimentally shown to outperform Gaussian-distribution-based kernel functions".

Presentation issues:

- It would be good to add figures to help illustrate the key concepts and proposed algorithms.

- The statement "On the other hand, $v^\top u_{max}$ can be computed beforehand during the indexing phase and can be easily accessed during the query phase" needs further clarification. How can this value be easily accessed given that $u_{max}$ depends on $q$?

- Typo: "with i.i.d. Gaussian entries)" => "with i.i.d. Gaussian entries."; "and Additional experimental results" => "and additional experimental results"

**Questions:**

There are no further questions.

---

> ### Author Response · Authors · 2025-11-20
>
> We thank the reviewer for acknowledging our contributions and giving insightful comments and suggestions. Our responses to the reviewer's questions are as follows.
>
> > (Weakness 1) Also, why are Tiny, GIST, and SIFT omitted from Table 1?
>
> We omitted these three datasets because they are image datasets and the distance metric is $l_2$ distance, while KS1 and CEOs are designed for maximum inner product search. The three datasets shown in Table 1 are text datasets evaluated using inner product.
>
> > (Weakness 2) It would be good to discuss if the proposed kernel functions are guaranteed to lead to more accurate (and/or more efficient) similarity search than Gaussian-distribution-based kernel functions given the same number of projection vectors. If this is not guaranteed, it would be good to tune down the claim: "can be both theoretically and experimentally shown to outperform Gaussian-distribution-based kernel function
>
> From the theoretical perspective, the new distribution can be guaranteed to be superior to the Gaussian distribution, because the reference angle of the new distribution is smaller, and a smaller reference angle leads to a more accurate estimation, as shown in Eq. 4 (please also see the analysis in lines 244–247).
>
> From the empirical perspective, the comparison between CEOs and KS1 was conducted under the same number of projection functions (see Table 1; 2048 denotes the number of projection vectors). The only difference between CEOs and KS1 is the distribution of the projection vectors. The result demonstrates that the proposed kernel functions lead to more accurate search.
>
> > (Presentation issues 1) It would be good to add figures to help illustrate the key concepts and proposed algorithms.
>
> We will add figures in the revised paper later.
>
> > (Presentation issues 2) The statement "On the other hand, $v^{\top}u_{\max}$ can be computed beforehand during the indexing phase and can be easily accessed during the query phase" needs further clarification. How can this value be easily accessed given that $u_{\max}$ depends on $q$?
>
> Although we do not know $u_{\max}$ in the indexing phase, we can store $v^{\top}u$ for every $u$. Then, in the query phase, after we know $u_{\max}$, we can access $v^{\top}u_{\max}$ immediately.
>
> We will add an explanation for this in the revised paper.
>
> > (Presentation issues 3) Typo: "with i.i.d. Gaussian entries)" => "with i.i.d. Gaussian entries."; "and Additional experimental results" => "and additional experimental results
>
> We will fix them in the revised paper.

---

> > ### Comment · Reviewer_XabQ · 2025-11-26
> >
> > Thank you for the response. My questions are all answered and I remain positive about the paper.

---

> > > ### Author Response · Authors · 2025-11-26
> > > **Re: Official Comment by Reviewer XabQ**
> > >
> > > Thanks for your positive feedback. We will revise the paper accordingly.

---

### Official Review · Reviewer_EhUf · 2025-11-02

**Soundness:** 3
**Presentation:** 3
**Contribution:** 3
**Rating:** 8
**Confidence:** 4

**Summary:**

The paper reframes angle estimation from stochastic Gaussian projections to deterministic reference-angle-based kernels. Authors first derive theoretical bounds and apply them to accelerating graph-based retrieval. Specifically authors design (what seems to be) a pruning algorithm that eliminates certain search paths. Although the pruning approach is not new in principle, the resulting approach HNSW-KS2 outperforms a previously proposed variant (HNSW-PEO) by 10% – 30%, along with a 5% reduction in index size.

**Strengths:**

1. Important topic and well motivated-problem

2. Theoretically grounded approach that achieves substantial practical gains.

3. The paper is well-written

4. Source code is provided.

**Weaknesses:**

1. Gains over HNSW-PEO are relatively modest (yet non-trivial!) and the method requires extra space, which is non-trivial in some cases. For example, it is >= 40% in the case of the SIFT dataset.
2. Evaluation is only single-threaded.


**Detailed comments:**

**Please, do not respond to these, all questions are rhetorical. If suggested correction is not valid, just ignore it**

Eq. (2) Shouldn’t Z_{HS} be Z_S?


L341 This is not understandable without a basic explanation of what a routing test is.

L410 there is a missing dot after and HNSW+PEOs

L427-428 On the other hand, in the high-recall region for Word, ScaNN outperforms HSNW+KS2 due to the connectivity issues of HNSW. -> This requires justification.

**Questions:**

**Detailed comments:**



1. Eq (1) does it really come from Theorem 3.1 in “Pham, Simple yet efficient algorithms for maximum inner product search via extreme order statistics.” It looks very different.

2. Due to its ease of implementation, CEOs has been employed in several similarity search tasks (Pham, 2021; Andoni et al., 2015; Xu & Pham, 2024) -> CEO came after Andoni et al. Do you mean it was used in FALCON++?

3. L354-356 you claim that you skip a distance computation. However, this doesn’t seem to be correct according to Alg 6. I think you also do not add a node to the queues. Basically, this seems to be search pruning approach, not the the approach to reduce the number of distance computations. Please, clarify.

4. L376 Why didn’t you test multi-threaded retrieval as well?



5. Is L the number of projections? It will be great to clarify in the experimental section.

6. Does Figure 1 compare PEO and KS2 using the same L?

7. Gains over CEO are marginal. For example (see Table 1), Probe@100: 6.98 vs 6.9, which is 1% relative. PEO improves upon reverse CEO and you improve upon PEO by double digits percentage points (e.g., 30%). How can you explain this discrepancy? What makes new kernel functions to be more effective? Is it due to the introduction of the threshold approximation? I think it is an important clarification to add to the paper as well.

8. Thank you for sharing the code: which part of the code benchmarks HNSW-PEO though? I think it benchmarks just HNSW-K1/K2.

---

> ### Author Response · Authors · 2025-11-20
>
> We thank the reviewer for acknowledging our contributions and giving insightful comments and suggestions. Our responses to the reviewer's questions are as follows.
>
> > (Comment 1) Eq (1) does it really come from Theorem 3.1 in “Pham, Simple yet efficient algorithms for maximum inner product search via extreme order statistics.” It looks very different.
>
> The requirement that $L$ is sufficiently large in Theorem 3.1 in the original paper is not accurate. Instead, this result  requires that $L$ goes to infinity; we have merely changed this expression. The other parts are kept unchanged.
>
> > (Comment 2) Due to its ease of implementation, CEOs has been employed in several similarity search tasks (Pham, 2021; Andoni et al., 2015; Xu & Pham, 2024) -> CEO came after Andoni et al. Do you mean it was used in FALCON++?
>
> Yes, it was used in FALCON++.
>
> > (Comment 3) L354-356 you claim that you skip a distance computation. However, this doesn’t seem to be correct according to Alg 6. I think you also do not add a node to the queues. Basically, this seems to be search pruning approach, not the the approach to reduce the number of distance computations. Please, clarify.
>
> In line 7 of Alg. 6, we test whether the node can pass the test. If not, this node will be discarded directly without executing the exact distance computation in line 8. This is the reason why we say the proposed test can reduce distance computations. In the original implementation, the distances from all nodes to the query need to be computed (line 8) to determine whether they can be added to the queue.
>
> > (Comment 4) L376 Why didn’t you test multi-threaded retrieval as well?
>
> Existing studies on CPU-based ANNS mainly  evaluated algorithm using a single thread for fair comparison. This is because (1) graph traversal for a single query is naturally not well-suited to multi-core execution, and (2) multi-core acceleration for multiple queries is always applicable. Notably, most graph-based ANNS algorithms, including the proposed method, can be accelerated using multiple cores, because searches for multiple queries are independent. For the above reasons, we chose single-thread evaluation for performance comparison, which corresponds to the contributions claimed in our paper.
>
> > (Comment 5) Is L the number of projections? It will be great to clarify in the experimental section.
>
> $L$ denotes the number of partitioned subspaces, which can be used to control the tradeoff between query accuracy and  efficiency.
>
> We will clarify the meaning of $L$ in the revised paper.
>
> > (Comment 6) Does Figure 1 compare PEO and KS2 using the same L?
>
> Yes, we use the same $L$ as in PEOs for every dataset.
>
> > (Comment 7) Gains over CEO are marginal. For example (see Table 1), Probe@100: 6.98 vs 6.9, which is 1% relative. PEO improves upon reverse CEO and you improve upon PEO by double digits percentage points (e.g., 30%). How can you explain this discrepancy? What makes new kernel functions to be more effective? Is it due to the introduction of the threshold approximation? I think it is an important clarification to add to the paper as well.
>
> First, we want to clarify that CEOs is compared with KS1 for MIPS, while reverse CEOs, PEOs, and KS2 need to be applied to similarity graphs for ANNS. Therefore, the gain of KS1 over CEOs and the gain of KS2 over PEOs/reverse CEOs are not directly comparable.
>
> Second, as for your question on where the effectiveness comes from, our answer is as follows.
>
> (1) The gain of KS1 over CEOs comes from only one difference: the distribution of the generated projected vectors, that is, Gaussian distribution vs. a more diverse distribution on the unit sphere, as we claim that the Gaussian distribution is suboptimal.
>
> (2) Except for the difference in distributions mentioned above, the gain of KS2 over PEOs comes from two additional aspects. In KS2, we use the information of the reference angle, which is not used in PEOs, and this improves the testing accuracy. On the other hand, the test function of KS2 is much simpler than that of PEOs, which accelerates the testing speed.
>
> For your suggestion on the above points, we will add a discussion in the revised paper.
>
>
> > (Comment 8) Thank you for sharing the code: which part of the code benchmarks HNSW-PEO though? I think it benchmarks just HNSW-K1/K2.
>
> We did not include HNSW-PEOs in our source code. We directly used the one provided by the authors of the PEOs paper, whose link can be found in the abstract of the PEOs paper.

---

> > ### Comment · Reviewer_EhUf · 2025-11-21
> > **pruning vs just reduction in distance-computation**
> >
> > >>(Comment 3) L354-356 you claim that you skip a distance computation. However, this doesn’t seem to be correct according to Alg 6. I think you also do not add a node to the queues. Basically, this seems to be search pruning approach, not the the approach to reduce the number of distance computations. Please, clarify.
> >
> > >In line 7 of Alg. 6, we test whether the node can pass the test. If not, this node will be discarded directly without executing the exact distance computation in line 8. This is the reason why we say the proposed test can reduce distance computations. In the original implementation, the distances from all nodes to the query need to be computed (line 8) to determine whether they can be added to the queue.
> >
> > I understand that you skip the distance computation. What I say: you **do more**. If read you algorithm correctly, the node is also not added to the queue. Isn't it? If this is the case, it's more than just skipping a distance computation. It's also pruning a search branch.

---

> > > ### Author Response · Authors · 2025-11-21
> > > **Re: pruning vs just reduction in distance-computation**
> > >
> > > Yes, you are right. That's the reason for the speedup. We will revise this part to make it clear.

---

### Author Response · Authors · 2025-12-03
**Author Remarks**

Dear Area Chair and Reviewers,

Thank you for your hard work in evaluating our paper.

**tl;dr**

**Contributions:** In this paper, we study efficient angle testing in the context of similarity search, and propose two projection-based probabilistic kernel functions which can be used to accelerate maximum inner product search (MIPS) and approximate nearest neighbor search (ANNS).

**Discussion Outcome:** All four reviewers gave high ratings (four 8's initially) and acknowledged our responses, maintaining their positive ratings during the discussion. All the reviewers responded by Nov. 26, 4am, UTC-12 (AoE), prior to OpenReview's system issue widely spread over the internet.

---

**Strengths Noted by the Reviewers:**

1. Important topic and well motivated-problem (Reviewers EhUf and XabQ);

2. Theoretically grounded approach (all four reviewers);

3. Methods integrated into existing algorithms such as HNSW (Reviewers yHFc and mXUD);

4. Practical gains demonstrated by experiments  (all four reviewers);

5. Source code provided (Reviewers EhUf and XabQ);

6. Well-written paper (Reviewer EhUf).

**Rebuttal and Revision Summary**

We answered all the questions raised by the reviewers. The paper was revised accordingly.

1. We refined the explanation of $L$ in lines 293–298. (Reviewer EhUf's Comment 5)

2. We added an explanation of why the proposed technique can outperform previous methods, in lines 362–367 for KS2 and lines 408–411 for KS1. (Reviewer EhUf's Comment 7)

3. We included a comparison with another related method, RabitQ (Gao et al., 2024), in lines 949–955. (Reviewer yHFc's Weakness 1)

4. We added an explanation of the probabilistic routing, which was proposed in the PEOs paper (Lu et al., 2024), in lines 960-1006. (Reviewer yHFc's Weakness 2)

5. We added an illustration (Fig. 3) showing the workflow of the KS2 test. (Reviewer XabQ's Presentation Issue 1)

6. We clarified the reason why the result in Lemma 1 is effective in lines 79–81. (Reviewer XabQ's Presentation Issue 2)

7. We fixed several typos in the paper.

Major revisions are marked in blue in the manuscript.

---

### Meta-Review · Area_Chair_FLEP · 2025-12-29

**Summary:**

The paper undertakes a study of angle estimation for similarity search and proposes theoretically grounded methods that yield empirical improvements. Reviewers were already initially unanimously positive about the paper. Authors provided comprehensive responses and clarifications in the rebuttal. The paper is recommended for acceptance.

**Reviewer Concerns:**

Reviewers voiced mild concerns about the overhead of the method, the extent of empirical gains, some aspects of evaluation setting, and presentation.

**Reviewer Scores:**

It turns that for this paper reviewer discussion with all reviewers had managed to take place before all discussions were halted, so there is little ambiguity. Scores would have very likely remained the same.

---

### Decision · Program_Chairs · 2026-01-26

Accept (Oral)